# On-chip integrated process-programmable sub-10 nm thick molecular devices switching between photomultiplication and memristive behaviour

Tianming Li[1,2,3], Martin Hantusch[4], Jiang Qu[1,2,3], Vineeth Kumar Bandari[1,2,3], Martin Knupfer[4], Feng Zhu [1,2,3,5✉] & Oliver G. Schmidt [1,2,3,6✉]

Molecular devices constructed by sub-10 nm thick molecular layers are promising candidates for a new generation of integratable nanoelectronic applications. Here, we report integrated molecular devices based on ultrathin copper phthalocyanine/fullerene hybrid layers with microtubular soft-contacts, which exhibit process-programmable functionality switching between photomultiplication and memristive behaviour. The local electric field at the interface between the polymer bottom electrode and the enclosed molecular channels modulates the ionic-electronic charge interaction and hence determines the transition of the device function. When ions are not driven into the molecular channels at a low interface electric field, photogenerated holes are trapped as electronic space charges, resulting in photomultiplication with a high external quantum efficiency. Once mobile ions are polarized and accumulated as ionic space charges in the molecular channels at a high interface electric field, the molecular devices show ferroelectric-like memristive switching with remarkable resistive ON/OFF and rectification ratios.

[1] Center for Materials, Architectures and Integration of Nanomembranes (MAIN), Chemnitz University of Technology, 09126 Chemnitz, Germany. [2] Material Systems for Nanoelectronics, Chemnitz University of Technology, 09107 Chemnitz, Germany. [3] Institute for Integrative Nanosciences, Leibniz IFW Dresden, 01069 Dresden, Germany. [4] Institute for Solid State Research, Leibniz IFW Dresden, 01069 Dresden, Germany. [5] State Key Laboratory of Polymer Physics and Chemistry, Changchun Institute of Applied Chemistry, Chinese Academy of Sciences, 130022 Changchun, China. [6] School of Science, Dresden University of Technology, 01069 Dresden, Germany. ✉email: zhufeng@ciac.ac.cn; oliver.schmidt@main.tu-chemnitz.de

Molecular-scale electronics refers to a domain where functional electrical circuits are created by individual or ensemble molecules[1]. It not only represents an attractive alternative route of miniaturization beyond Moore's law for future electronic devices with low energy consumption and rich functionalities, but also provides an ideal platform to explore the multiple properties of molecules under mixed external stimuli, thereby extending the conventional mechanics, optoelectronics, thermoelectrics, and spintronics to the molecular level[2]. At the same time, it is believed that the coupling between ionic and electronic charges, which has already played essential roles in many mesoscopic devices (such as ion batteries[3], ionic actuators[4], and electrochemical transistors[5]), could further expand the functional diversity of molecular devices, because the coexistence of electrons, holes, anions, and cations in molecularly thin channels might open up additional possibilities for the molecules to react with various stimuli[6]. Apart from that, the chemical activity of the ions can provide even more functionalities to the molecular devices. Recently, Han et al. demonstrated an electric-field-driven molecular switch based on a self-assembled monolayer (SAM) and a GaO$x$/EGaIn liquid top electrode[7], which formed a molecular junction with the dual functionality of a diode and a variable resistor. The authors ascribed this phenomenon to the dimerization of redox units, resulting in the hybridization of molecular orbitals accompanied by directional migration of the counterions.

Generally, molecular devices focus on molecular junctions, in which the functionalities are realized by manipulating the chemical properties and electronic structures of the individual or ensemble molecules[8,9]. However, the potential of molecular devices constructed by ultrathin molecular layers with dimensions comparable to ensemble molecular junctions has been underestimated. For instance, fundamental studies predicted that short-channel molecular rectifiers can achieve rectification frequencies in the terahertz range[10,11], but a state-of-the-art half-wave SAM rectifier (consisting of Ag$^{TS}$-SC$_{11}$Fc$_2$//Ga$_2$O$_3$/EGaIn) was reported to operate at only 50 Hz[12]. In marked contrast, Li et al. constructed integrated rectifiers based on nanometer-thick hybrid layers consisting of fluorinated cobalt phthalocyanine (F$_{16}$CoPc)/copper phthalocyanine (CuPc) heterojunctions, which were able to convert alternating current to direct current with a frequency of up to 10 MHz[13]. The excellent high-frequency response was mainly attributed to the band bending of the ultrathin heterojunction. The same dimension but different working principle attracts our special interest to further integrate and control the ionic and electronic functions within ultrathin molecular layer-based devices on the wafer-scale, which is crucial to develop novel and practical functional molecular devices.

To achieve this goal at least two prerequisites must be met: a reliable route for fabricating sub-10 nm thick molecular devices and a convenient way to bring mobile ions into the molecular channels. Molecular junctions are the most intensively investigated sub-10 nm thick molecular devices. Compared to single-molecule junctions, large-area junctions based on molecular layers are more promising for scalable fabrication. Various fascinating techniques have been developed to realize ensemble molecular junctions on large scales, such as polydimethylsiloxane (PDMS)-assisted liquid metal contacts[14], suspended nanowire junctions[15], surface-diffusion-mediated deposition[16], nanoskiving[17], carbon paint protective layer-based junctions[18], and electron-beam deposition of carbon[19]. A comparison of these techniques for large-area sub-10 nm thick molecular devices is discussed in Supplementary Note 1 and Supplementary Table 1. Among them, rolled-up nanotechnology provides an efficient strategy to fabricate fully integrated functional molecular devices on chip via damage-free soft contacts[20], which is not only suitable for ordered self-assembled monolayers but also for deposited ultrathin molecular layers. The latter allows for great flexibility in designing the molecular component and thickness through standard vacuum deposition technologies such as thermal evaporation[13,21]. Regarding the mobile ions to be inserted into the molecular channels, poly(3,4-ethylenedioxythiophene):polystyrene sulfonate (PEDOT:PSS), a mixed ionic-electronic polymer conductor, is selected because it not only serves as a transparent electrode but also provides an "ion reservoir" to supply mobile ions[22].

Enabled by the nondestructive rolled-up top electrodes, we realize on-chip integrated process-programmable molecular devices by controlling electronic charges and mobile ions in the active channels at the molecular level. Based on this configuration, the molecular devices are able to switch between photomultiplication photodiodes and bipolar memristors under different external stimuli. The schematic structure is shown in Fig. 1. By modulating the interfacial electric field between the PEDOT:PSS electrode and the enclosed molecules via tuning the carrier injection barrier, the release of ions stored in the PEDOT:PSS matrix can be controlled, which in turn provides molecular devices with ferroelectric-like memristive switching and photomultiplication functionalities. The function switching therefore originates from the synergetic ionic-electronic transport and is driven by the applied electric field. At a low interfacial electric field, electronic transport dominates and multiplied photocurrents flow through the sub-10 nm thick molecular devices with ultrahigh external quantum efficiency (EQE, up to $10^4$%) at a low applied voltage (down to 1 V). At a high interfacial electric field, ionic migration is activated and the molecular devices act as bipolar memristive switches with both remarkable resistive ON/OFF and rectification ratios.

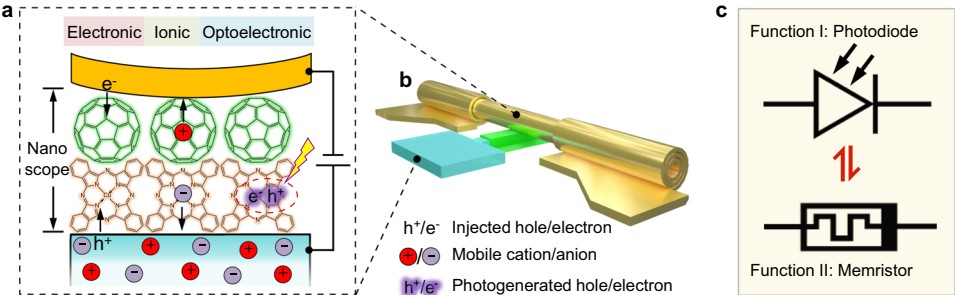

**Fig. 1 Schematic showing the concept of functional molecular devices based on electronic-ionic synergistic effects. a** Schematic illustration of electronic/ionic/electronic-ionic transport on the molecular scale under various external stimuli such as electrical and optical excitation. One electrode (e.g., the bottom electrode) also acts as a controllable "ion reservoir" to supply mobile ions. **b** The concept in **a** can be realized by integrated molecular devices based on damage-free rolled-up soft contacts. **c** Based on the configuration in **b**, a transition between photomultiplication and memristive switching is realized in the on-chip integrated devices by controlling the electronic-ionic transport in the molecule channel.

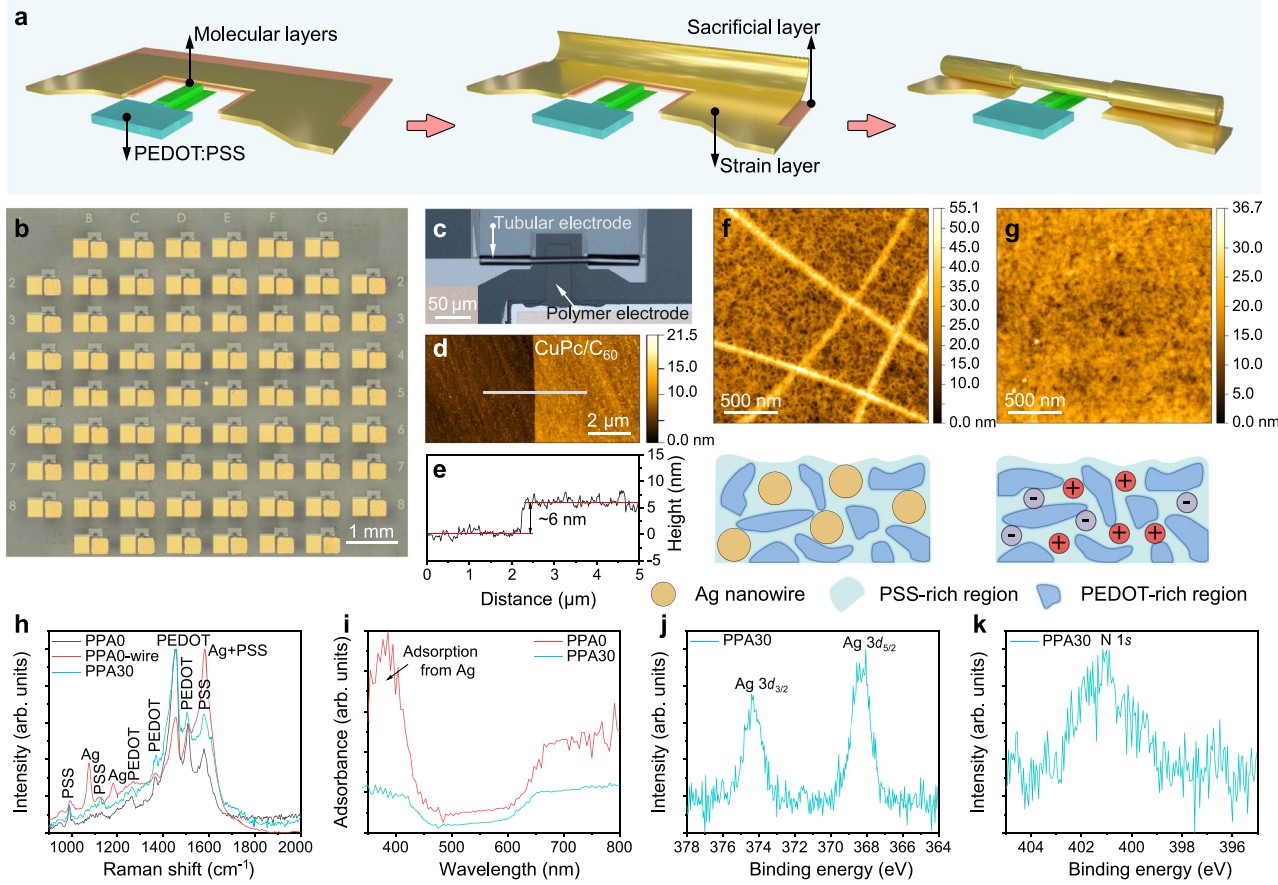

**Fig. 2 Design and microfabrication of the molecular devices. a** Formation of rolled-up soft contacts. The finger-shaped mesa structure provides a platform for the bottom electrode, onto which molecular layers are grown. During the selective etching of the sacrificial layer GeOx by deionized (DI) water, the Au/Ti/Cr nanomembranes release their built-in stress and roll up. After rolling, the rolled-up metallic nanomembrane bonds as a top electrode to the molecule layer below, forming a sandwich structure. **b** Micrograph of the molecular device array. **c–e** Typical single device based on a rolled-up soft contact (**c**). The width of the bottom electrode and the diameter of the rolled-up metallic tube are ~40 μm and ~10 μm, respectively. AFM image of CuPc (3 nm)/CuPc (3 nm) grown on the wafer (**d**) and the corresponding height profile (**e**). **f, g** AFM images of the spin-coated PEDOT:PSS:AgNWs film before (**f**) and after (**g**) 30 s wet etching. Schematic illustrations are also given. **h, i** Raman spectrum (**h**) and adsorption coefficient profile (**i**) of the spin-coated PEDOT:PSS:AgNWs film before (denoted as PPA0) and after 30 s wet-etching (denoted as PPA30). **j, k** XPS spectra (Al-Kα = 1486.6 eV) corresponding to Ag 3d (**j**) and N 1s (**k**) of the etched PEDOT:PSS:AgNWs film.

## Results

**Design and microfabrication of the molecular devices**. Rolled-up metallic nanomembranes formed by releasing built-in stress are adopted as a damage-free and self-adjusted top electrode array to the molecular layers in this work (Fig. 2a). A more detailed description of the fabrication process is provided in Supplementary Note 2 and Supplementary Fig. 1. Figure 2b shows a microscope image of the device array. The fabrication process is based on standard photolithography and vacuum deposition technologies, which are the two key fabrication technologies for modern integrated circuits (ICs). By the rolled-up soft contacts, molecular hybrid CuPc/$C_{60}$ layers with various nanometer-level thicknesses are sandwiched between the finger-like bottom electrodes and tubular top electrodes, as shown in Fig. 2c. The apparent height of the nominal CuPc (3 nm)/$C_{60}$ (3 nm) hybrid layer is measured to be 6 nm (Fig. 2d, e), where CuPc is calibrated to be 2.9 nm and $C_{60}$ to be 3.1 nm (Supplementary Note 3 and Supplementary Fig. 2). Since the deviation is small, we still use the nominal height to label the molecular devices. It is reasonable to claim that the CuPc/$C_{60}$ molecular layer falls into the sub-10 nm thickness scale.

The solution consisting of poly(3,4-ethylenedioxythiophene):-polystyrene sulfonate and silver nanowires (denoted as PEDOT:PSS:AgNWs) was spin-coated on the glass wafer and then patterned as transparent bottom electrode arrays that cover the finger-shaped mesa structures. To assist the pattern transfer process, a silver protecting layer (200 nm) was deposited on the PEDOT:PSS:AgNWs polymer film, followed by acid wet etching and $O_2$ plasma etching (Supplementary Note 4 and Supplementary Fig. 3). The wet etching realized by the mixed acid solution ($H_3PO_4$:$HNO_3$:$CH_3COOH$:$H_2O$) removes not only the Ag protecting layer but also the AgNWs inside the PEDOT:PSS matrix, as shown in Fig. 2f, g, and Supplementary Fig. 4. The etched PEDOT:PSS:AgNWs film is characterized as an excellent transparent electrode material, which is confirmed by its high electrical conductivity and good light transmission (Supplementary Note 5, Supplementary Figs. 5, 6). On the other hand, based on the Raman spectrum and absorbance profile, no Ag metal signal is detected after etching (Fig. 2h, i). However, the XPS spectrum exhibits a clear Ag signal (Fig. 2j and Supplementary Table 2). According to the literature using the same PEDOT:PSS:AgNWs[23], Auger spectroscopy revealed that Ag within the sampling depth occurs in the form of Ag ions (usually 2–5 nm). At the same time, an N 1s peak (Fig. 2k) is observed, which originates from the etching component nitric acid ($HNO_3$). Based on the above observations, we conclude that the etched

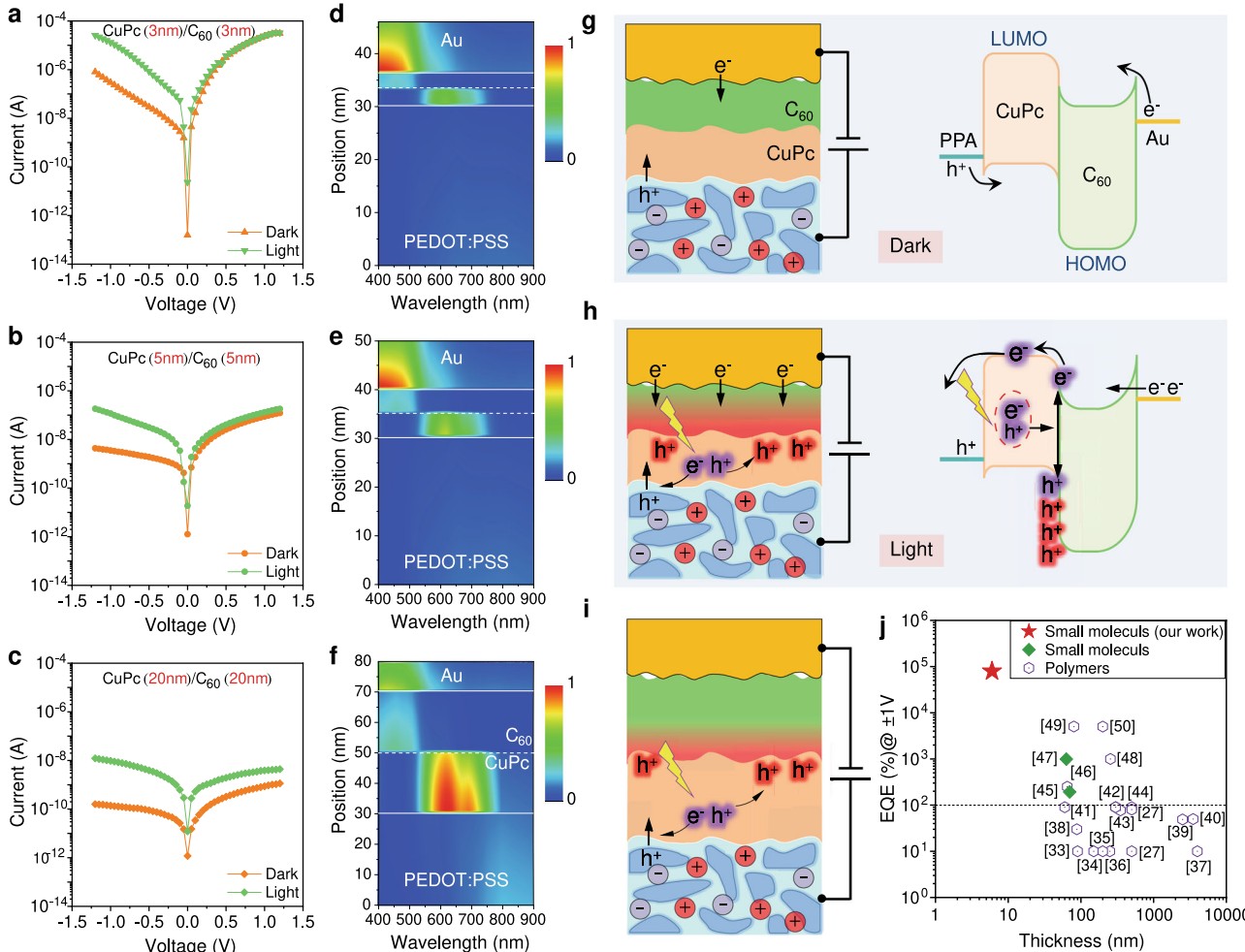

**Fig. 3 Molecular photomultiplication photodiodes based on PPA30/CuPc (x nm)/C$_{60}$ (x nm)/Au tube. a–c** I-V characteristics of the molecular devices in the dark and under light illumination (**a** x = 3 nm; **b** x = 5 nm; **c** x = 20 nm). The bottom electrode is grounded during the I-V measurement. Both the dark and photo currents increase with the decrease in thickness, while the photo/dark current ratios remain close to 100. **d–f** Simulation of photogenerated exciton distribution in the molecular devices (**d** x = 3 nm; **e** x = 5 nm; **f** x = 20 nm). **g, h** Schematic showing the molecular devices with an ultrathin molecular hybrid layer in the dark (**g**) and under illumination (**h**). Purple-glowing "e$^-$" and "h$^+$" denote photogenerated free electrons and holes, respectively. Trapped photogenerated holes by the C$_{60}$ layer are marked as red-glowing "h$^+$". The red gradient area represents the region affected by the trapped photogenerated holes. **i** Schematic illustration of the molecular devices with a thick molecular hybrid layer under illumination, in which the effect of trapped photogenerated holes cannot reach the top Au electrode, because the trapped holes have a low density and are located relatively far away from the Au tube electrode. **j** Comparison of EQE (at 1 V or −1 V) of our photomultiplication photodiodes (PPA30/CuPc (3 nm)/C$_{60}$ (3 nm)/Au tube to 630 nm laser light with a power density of 5 mW cm$^{-2}$) with previously reported results that are based on polymer/organic active materials[27,33–50].

PEDOT:PSS:AgNWs film can serve as a good transparent electrode and ion reservoir.

**Molecular photomultiplication photodiodes**. When the PEDOT:PSS:AgNWs/Ag film (200 nm) is etched in the acid solution for 30 s, the treated bottom electrode is labelled as PPA30 for convenience. And the corresponding molecular devices based on the CuPc (x nm)/C$_{60}$ (x nm) hybrid layer are denoted as PPA30/CuPc (x nm)/C$_{60}$ (x nm)/Au tube, where x = 3, 5, 20. Figure 3a–c show the I-V curves of the devices in the dark and under white light (100 mW cm$^{-2}$) illumination. The devices consist of CuPc (3 nm)/ C$_{60}$ (3 nm), CuPc (5 nm)/C$_{60}$ (5 nm), and CuPc (20 nm)/C$_{60}$ (20 nm), respectively, and all of them exhibit remarkable photoresponse similar to a photodiode. The function of a photodiode is to convert light into a current signal. One important index is the external quantum efficiency (EQE)[24], which is defined as the ratio of the converted electrons to the incident photons. According to this definition the EQE values of the devices based on CuPc (3 nm)/C$_{60}$

(3 nm), CuPc (5 nm)/C$_{60}$ (5 nm), and CuPc (20 nm)/C$_{60}$ (20 nm) at −1 V are roughly calculated to be greater than 33193%, 293%, and 24.5%, respectively, when taking the maximum wavelength of white light (i.e., 700 nm) as the incident light. It is interesting that devices based on (3 nm)/C$_{60}$ (3 nm) and CuPc (5 nm)/C$_{60}$ (5 nm) exhibit an EQE exceeding 100%, thus falling into the category of photomultiplication photodiodes[25] which are different from the traditional photodiodes that rely on the increase in carrier density under illumination (i.e., EQE < 100%). A brief comparison between a photomultiplication photodiode and a traditional photodiode is provided in Supplementary Note 6 and Supplementary Fig. 7. The response to single-wavelength light further confirms the photomultiplication behaviour, as shown in Supplementary Note 7 and Supplementary Fig. 8. The EQE values (at −1 V) of the PPA30/CuPc (3 nm)/C$_{60}$ (3 nm)/Au tube in response to light (5 mW cm$^{-2}$) with a wavelength of 459, 532, and 630 nm are 21124%, 29051%, and 79241%, respectively.

A photomultiplication photodiode relies on the improved charge injection induced by trapping the photogenerated carriers

of opposite sign in the trap states near the electrode, which can produce a strong local Coulomb field and thus narrow the barrier width[26,27]. Normally, the voltage values in previous reports are approximately tens of volts, because the thick photosensitive channels (hundreds to thousands of nanometers thick) need a very high electric field strength to bring a sufficient number of photogenerated carriers close to the electrodes[26]. However, in this work (as shown in Fig. 3a), the applied voltage for photo-multiplication is greatly reduced to ~1 V owing to the ultrathin organic heterojunction. At only −0.1 V, the calculated EQE achieves 286% (Supplementary Fig. 9a). However, the response and recovery processes occur on a time scale of seconds (Supplementary Fig. 9b), which are much slower than traditional photodiodes (in the nanosecond to millisecond range). The slow photoresponse of this integrated molecular device is consistent with the feature of photomultiplication, mainly resulting from the slow trapping/detrapping process[28].

The depth profiles of the wavelength-dependent light absorption in PPA30/CuPc ($x$ nm)/$C_{60}$ ($x$ nm)/Au tube are simulated in Fig. 3d–f. They reveal that the CuPc film is the main photosensitive material and 20 nm thick CuPc film is within the effective adsorption depth. In view of the band structure (Fig. 3g), the PPA30/CuPc ($x$ nm)/$C_{60}$ ($x$ nm)/Au tube devices can be simplified as back-to-back diodes connected in an anti-parallel configuration, where the junction barriers of the back-to-back diodes arise from the CuPc/$C_{60}$ and $C_{60}$/Au tube, respectively. Considering the $I$-$V$ rectifying direction in the dark (as shown in Fig. 3a–c), the Schottky junction between $C_{60}$/Au is dominant over the p-n junction between CuPc/$C_{60}$. Therefore, there is a barrier for electron transport between the lowest unoccupied molecular orbital (LUMO, about 3.6 eV[29,30]) of n-type $C_{60}$ and the Fermi level ($E_F$) of the Au tubular electrode. According to our previous work, the work function of the "Au tube" was measured to be about 4.25 eV[13], which can theoretically create a barrier height of ~0.65 eV between $C_{60}$ and the Au tube. Without illumination, the relatively low dark current is due to the large charge injection barrier under reverse bias (the etched PEDOT:PSS:AgNWs bottom electrode is grounded), as illustrated in Fig. 3g. While under illumination, photon absorption in CuPc or $C_{60}$ molecules creates excitons which quickly dissociate at the CuPc/$C_{60}$ interface, hence generating holes on the highest occupied molecular orbital (HOMO) of CuPc and electrons on the lowest unoccupied molecular orbital (LUMO) of $C_{60}$. When a reverse bias is applied, the photogenerated electrons and holes move along the potential gradient, and the holes are trapped by $C_{60}$ due to the large HOMO level offset (theoretically about 1 eV) at the CuPc/$C_{60}$ interface[29]. When the $C_{60}$ layer is as thin as the "interface", the trapped holes are equivalently accumulated in the vicinity of the Au tube electrode, thereby producing a high local Coulomb field[31] across the entire $C_{60}$ ultrathin film. The local high electric field causes interfacial band bending (i.e., narrowing the Schottky barrier) which facilitates electron tunnelling injection from the Au tube to the $C_{60}$ molecules. Consequently, the electrons can be easily injected with a small reverse bias, resulting in a multiplied photocurrent (as shown in Fig. 3h). In other words, the photomultiplication originates from the enhanced electron injection induced by trapping photogenerated holes in the hole-blocking layer (i.e., the $C_{60}$ layer). The EQE of CuPc (20 nm)/$C_{60}$ (20 nm) is much smaller than 100%, because the larger layer thickness leads to fewer excitons reaching the CuPc/$C_{60}$ interface. Therefore, the trapped holes have a low density, and are localized far away from the Au tube electrode, which cannot generate a strong electric field to induce significant band bending at the $C_{60}$/Au interface, as shown in Fig. 3i. In this case, the photoelectric effect-induced carrier density increase dominates, which coincides with the fact that only the devices

based on CuPc (20 nm)/$C_{60}$ (20 nm) exhibit an obvious photoresponse at forward bias (Fig. 3a–c). Therefore, there is a pronounced thickness-dependent photomultiplication effect. This conclusion is further confirmed by transition voltage spectroscopy (TVS)[32] as presented in Supplementary Note 8 and Supplementary Fig. 10.

Compared to previous reports on molecule/polymer-based photomultiplication photodiodes (Fig. 3j)[27,33–50], we fabricated molecular photomultiplication photodiodes with only several nanometer-thick molecular layers, a very low working voltage down to 1 V, and an ultrahigh EQE (at the level of $10^4$%).

**Molecular bipolar memristors.** The photomultiplication photo-diodes are dominated by electronic transport. As mentioned before, the etched PEDOT:PSS:AgNWs film contains ions and can act as an ion reservoir to supply mobile ions. Therefore, several questions arise: What happens if ions join transport? And how can ionic transport be triggered? We find that when the wet etching time of the PEDOT:PSS:AgNWs/Ag film (200 nm) is extended to 60 s (labelled as PPA60), the corresponding molecular devices PPA60/CuPc ($x$ nm)/$C_{60}$ ($x$ nm)/Au tube can exhibit $I$-$V$ hysteresis loops over a certain range of applied voltages, as shown in Fig. 4a–f. Taking the PPA60/CuPc (3 nm)/$C_{60}$ (3 nm)/Au tube as an example (Fig. 4a, b), with a small applied voltage there is nearly no $I$-$V$ hysteresis (Fig. 4a). However, when a higher positive sweep voltage is applied (0 V→1.2 V→0 V), the current increases sharply, resulting in an $I$-$V$ hysteresis loop (Fig. 4b). Bidirectional switching is observed because the application of a negative sweep voltage (0 V→−1.2 V→0 V) also generates a loop. The resistive ON/OFF ratio at −0.5 V is about 150, while the ratio at 0.5 V is about 1500. According to literatures[51,52], the molecular devices based on PPA60/CuPc (3 nm)/$C_{60}$ (3 nm)/Au tube demonstrate ferroelectric-like memristive switching performance. PPA60/CuPc (5 nm)/$C_{60}$ (5 nm)/Au tube exhibits a similar phenomenon but with a higher driving voltage to trigger the $I$-$V$ hysteresis (Fig. 4c, d), which occurs during 0 V→2.2 V→0 V→-2.2 V→0 V. However, the devices consisting of CuPc (20 nm)/$C_{60}$ (20 nm) show a very weak hysteretic effect even when the applied voltage is ramped up to 5.2 V (Fig. 4e, f). The thickness-dependent performance reveals that the thinner the molecular layer is, the easier the memristive switching occurs for the device in this work. Statistics of yield and resistance switching performance are provided in Supplementary Note 9 and Supplementary Fig. 11.

The resistance switching in the PPA60/CuPc/$C_{60}$/Au tube structure is further explored. As described in detail in Supplementary Note 10, the $I$-$V$ hysteresis order and linear fitting (Supplementary Fig. 12), the switchable diode effect (Supplementary Fig. 13), and the hysteresis reduction during light illumination (Supplementary Fig. 14) suggest that the hysteretic behaviour of our molecular devices is analogous to that of barrier modulation-related perovskite memristors, instead of conventional filamentary memristors. For memristors based on perovskite materials, e.g., $BiFeO_3$ films with defects such as bismuth and oxygen deficiencies (acting as mobile ions)[52,53], the hysteresis is attributed to the Schottky barrier modulation induced by ion polarization in the perovskite storage medium. A brief introduction to ion accumulation induced modulation of charge injection is provided in Supplementary Note 11 and Supplementary Fig. 15. For perovskite materials, the mobile ions originate from the undercoordinated ionic defects at the surface and grain boundaries[54]. However, in our work a series of control experiments confirm that the mobile ions come from the etched PEDOT:PSS:AgNWs film rather than the active materials, i.e., CuPc/$C_{60}$ (Supplementary Note 12, Supplementary Figs. 16, 17).

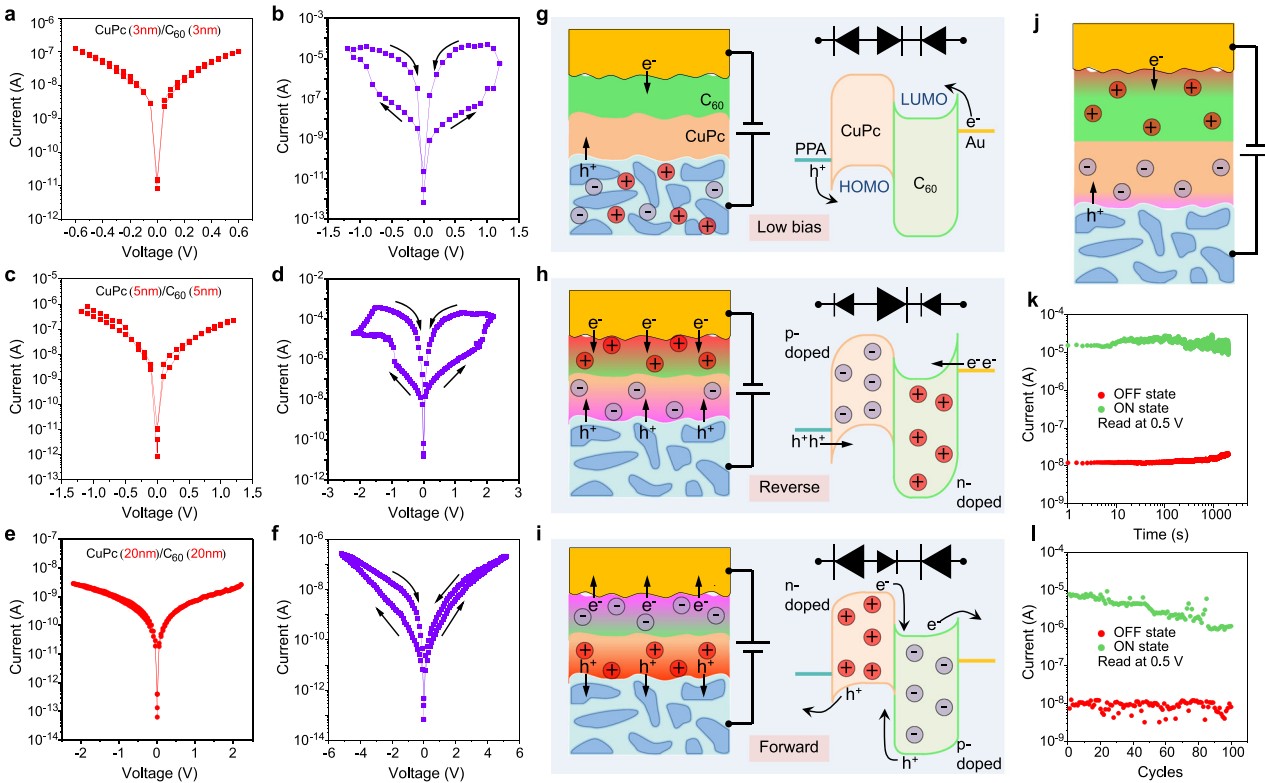

**Fig. 4 Molecular bipolar memristors based on PPA60/CuPc ($x$ nm)/$C_{60}$ ($x$ nm)/Au tube. a–f** $I$-$V$ characteristics of the molecular devices in different voltage windows ((**a**) and **b**, $x = 3$ nm; **c** and **d**, $x = 5$ nm; **e** and **f**, $x = 20$ nm). In the low applied voltage range, the molecular devices show nearly symmetric $I$-$V$ curves without a loop. At higher voltages electric hysteresis loops appear. **g–i** Schematic showing the molecular devices with an ultrathin molecular hybrid layer (**g** low bias; **h** high reverse bias; **i** high forward bias). The red gradient area represents the n-doped region induced by cation accumulation, while the purple gradient area represents the p-doped region induced by anion accumulation. When firstly biased by a high reverse voltage, ion polarization makes the system similar to a diode dominated by the p-n junction (**h** and Supplementary Fig. 13d). When firstly biased by a high forward voltage, the system can be considered as a diode dominated by the two Schottky junctions (**i** and Supplementary Fig. 13e). **j** Schematic illustration of the molecular devices with thick molecular hybrid layers under reverse bias, in which the accumulated ions have a much weaker modulation effect for carrier injection due to the ineffective ion polarization and relatively low ion concentration. **k** Retention of ON and OFF states of PPA60/CuPc(3 nm)/$C_{60}$(3 nm)/ Au under continuous read-out voltages (at 0.5 V). The device is switched to the ON state by applying a positive voltage pulse at 1.5 V for a few seconds. **l** Endurance of ON and OFF states of PPA60/CuPc(3 nm)/$C_{60}$(3 nm)/Au over the first 100 measurement cycles (read at 0.5 V).

This is the first time to construct integrated molecular devices in which mobile ions initially residing in the electrode are brought into the active molecular channel. According to Refs. [55,56], ion intake can decrease the work function of PEDOT:PSS, hence, the PPA60/CuPc ($x$ nm)/$C_{60}$ ($x$ nm)/Au tube system can be treated as three diodes in series, which comprise two Schottky junctions and one p-n junction (as shown in Fig. 4g). By increasing the reverse voltage to a sufficiently high value, positive cations escape from the PEDOT:PSS:AgNWs electrode and accumulate at the $C_{60}$/Au tube interfacial region, further n-doping the n-type $C_{60}$ layer near the Au electrode[52,57]. The cation accumulation can give rise to a large electric field at the interface, which reduces the barrier width for electron injection from the Au tube electrode. At the same time, the remaining negative anions accumulate in the vicinity of the PPA60 surface, further p-doping the nearby p-type CuPc layer, thereby enhancing the hole injection from the PPA60 electrode. When the hybrid layer is ultrathin, for example, CuPc (3 nm)/$C_{60}$ (3 nm), the molecular layer can be considered as the "interface" and can be fully doped. This can not only narrow the junction barrier between the electrodes and the enclosed molecules to improve electron/hole injection, but also increase the built-in potential at the CuPc/$C_{60}$ p-n junction, as shown in Fig. 4h. In this situation, PPA60/CuPc (3 nm)/$C_{60}$ (3 nm)/Au tube under reverse bias is equivalent to a diode dominated by the strengthened p-n junction. Similarly, for the forward-biased

condition, the p-type CuPc layer is n-doped while the n-type $C_{60}$ layer is p-doped. Consequently, the Fermi level offset between CuPc and $C_{60}$ could be reduced, leading to a decrease in the built-in potential, while the two electrode-related barriers are increased, as shown in Fig. 4i. Therefore, the system under forward bias can be considered as a diode dominated by the two enhanced Schottky junctions. In summary, the electric-field-driven ionic doping contributes to both the resistive switching and switchable rectifying phenomena.

The voltage- and thickness-dependent switching performance (as shown in Fig. 4a–f) can be well explained by ionic polarization induced modulation. Specifically, an appropriately high voltage is required to activate the transport of ions, leading to ion polarization and the corresponding memristive effect. This is why there is nearly no hysteresis in the $I$-$V$ curve with a relatively small applied voltage. On the other hand, when the thickness of the CuPc/$C_{60}$ molecule layer is increased, the applied voltage triggering the hysteresis also increases. This is because more potential drops over the thicker hybrid layer, resulting in less potential drop across the PPA60/CuPc interface. In other words, the interfacial local electric field decreases with a thicker active layer, which makes it harder to trigger ion transport and a higher voltage for activation is required. According to literature[58], the location, concentration and distribution of ions in the active layers can control the conductance, rectification and switching

polarity of the device. When the molecular hybrid layer is too thick, the motion of mobile ions encounters continuous obstruction through the bulk molecular layer which impedes effective ion polarization. Furthermore, the relative ion concentration is lower within a thicker active layer. Both effects reduce the impact of the ions, resulting in a minor resistive switching as shown in Fig. 4f, j.

The stability of our molecular memristors was characterized. The retention time of the PPA60/CuPc(3 nm)/C$_{60}$(3 nm)/Au device is stable and reliable over 2000 s, and the ON/OFF current ratio remains at ~$10^3$ with a continuous reading bias of 0.5 V, indicating good retention stability (Fig. 4k). The endurance of the molecular device degrades slightly over the first 100 sweep cycles due to the decrease in the ON-state current, however, the ON/OFF ratio remains in the range of $10^2$ (Fig. 4l). Compared with previously reported bipolar memristors (Supplementary Note 13 and Supplementary Table 3), the key parameters (such as storage medium thickness, rectification ratio, ON/OFF ratio, switching voltage, ON-state current density, and so on) of our molecular memristors reach a comparable level except for the endurance stability.

**Mechanism of the electric-field-driven transition.** Tuning the barrier through doping near the contacts ($\omega \propto 1/\sqrt{N_D}$, where $\omega$ and $N_D$ are the barrier width and dopant density, respectively[59]) is a standard practice in field-effect transistors (electronic doping)[60] and light-emitting electrochemical cells (ionic doping)[61]. For the nanoscale molecular devices in this work, the memristive phenomenon is attributed to the ionic doping induced by ion polarization, while the photomultiplication effect realized by photogenerated-carrier trapping can be regarded as the result of electronic doping, both of which can reduce the barrier widths for carrier injection, as shown in Fig. 5a. Therefore, the effects of trapped holes and accumulated ions in the vicinity of the electrode have a similar effect, i.e., increasing the local carrier density ($N_D$). However, how do these two distinct phenomena happen with the same PEDOT:PSS:AgNWs/CuPc/C$_{60}$/Au tube configuration? The photoelectron spectra reveal that extending the wet-etching time (from 30 s to 60 s) decreases the work function of PEDOT:PSS:AgNWs from 4.8 eV to 4.5 eV, as shown in Fig. 5b (middle). At the same time, the relative atomic content of Ag 3$d$ increases (Supplementary Table 2). According to literatures[55,56], ion intake can decrease the work function of PEDOT:PSS, which is consistent with our result. Under this circumstance, the barrier height between the Fermi level (E$_F$) of the PPA electrode and the HOMO of the CuPc molecules increases by ~0.3 eV, thereby making the contact PPA60/CuPc contact Schottky-like and PPA30/CuPc ohmic-like (or weak Schottky-like), as shown in Fig. 5b. At a given applied voltage, the potential drop over the contact increases when the barrier height increases, leading to a higher local electric field at the PPA60/CuPc interface than that at the PPA30/CuPc interface. It is well known that the activation energy of ion transport is field-dependent and decreases with the square of the applied voltage[62]. In other words, due to the higher interfacial electric field, ion transport in molecular devices based on the PPA60 electrode is much easier to be activated than that based on PPA30. This can reasonably explain the transition between the electronic transport-dominated photomultiplication photodiode and the synergetic electronic-ionic transport-dominated bipolar memristor (Fig. 5c), which is controlled by the accumulation of photogenerated holes or mobile ions. Another experimental evidence is that when increasing the water washing time (from 10 s to 10 min) of PPA60 after wet-etching (i.e., before molecule deposition), the corresponding molecular devices act as photomultiplication

photodiodes again, rather than bipolar memristors (Supplementary Fig. 17). This is because the longer washing procedure partly desorbs the mobile ions, leading to a decrease in ion concentration and a reduction of the PPA60/CuPc barrier height, both of which make the ion transport difficult. Apart from tuning the junction barrier, the interfacial local electric field can also be increased by directly increasing the applied voltage, as shown in Fig. 4a, b.

## Discussion

In conclusion, we developed integrated process-programmable molecular devices by modulating the coupling between electronic and ionic carriers in sub-10 nm molecular-scale channels. The integrated molecular devices demonstrate an electric-field-driven transition between photomultiplication photodiodes and bipolar memristors, which is controlled by the release of mobile ions stored in the PEDOT:PSS bottom electrode. At a low interfacial electric field, ions are not driven into the molecular channels and remain in the PEDOT:PSS electrode, thus, electronic charge transport dominates the device performance. Under illumination, the photogenerated holes are trapped by the ultrathin hole blocking layer near the top electrode, which induces interfacial band bending to assist the tunnelling injection of electrons from the top electrode, resulting in photomultiplication with ultrahigh external quantum efficiency at low applied voltage. At a high interfacial electric field ionic migration is activated. The polarized mobile ions are accumulated in the vicinity of the bottom and top electrodes, and can modify the interfacial barrier as well as the electronic carrier injection, resulting in ferroelectric-like memristive switching with both a remarkable resistive ON/OFF ratio and rectification ratio. Therefore, both trapped photogenerated carriers and accumulated mobile ions have the function of modulating the carrier injection from an external circuit by inducing band bending at the interface, which is highly effective when the molecular layers are as thin as the "interface". For the first time, the coupling of electronic and ionic charge processes is employed to modulate functionalities in integrated molecular devices. The combination of the "soft-contact" enabled by rolled-up nanotechnology and the "ion reservoir" provided by the polymeric electrode opens up a novel strategy for generating a vast range of functional molecular devices based on the synergistic electronic-ionic reaction to various stimuli.

## Methods

**Materials.** Photoresist AR P5910 purchased from Allresist GmbH was applied to etch the mesa structures. Photoresist AZ 5214E purchased from Microchemicals GmbH was used for lift-off processes. Clevios™ HY E (PEDOT:PSS:AgNWs) was used as the bottom electrode material. Molecular materials copper (II) phthalocyanine (CuPc, >99%) and fullerene (C60, >99%) were purchased from Sigma-Aldrich without any purification.

**Fabrication.** Rolled-up nanotechnology was applied to assemble the molecular devices on the wafer scale via conventional photolithography techniques, in which the strained metallic nanomembranes released the stress and automatically rolled up as the damage-free top tubular electrodes to molecules. Conductive PEDOT:PSS:AgNWs were patterned as the transparent bottom electrode and ion reservoir by Ag film assisted pattern transfer. Molecules were deposited by using low temperature evaporation in vacuum with a base pressure of $10^{-7}$ mbar. A detailed description of the fabrication process is provided in Supplementary Note 2 and 4.

**Characterization.** The topography images and the height profiles were obtained by a Bruker atomic force microscope with tapping mode. Raman spectroscopy was carried out at room temperature by a LabRAM HR Evolution (Horiba). The adsorption coefficient was calculated with a light source provided by QuantX 300 which was monitored by a laser power meter PM110D (THORLABS GmbH). A monochromatic Al Kα source with an energy of 1486.6 eV was used to collect the surface information of the samples about the core-levels and valence levels. During photoelectron spectroscopy acquisition, samples were under a bias of 5 V to obtain

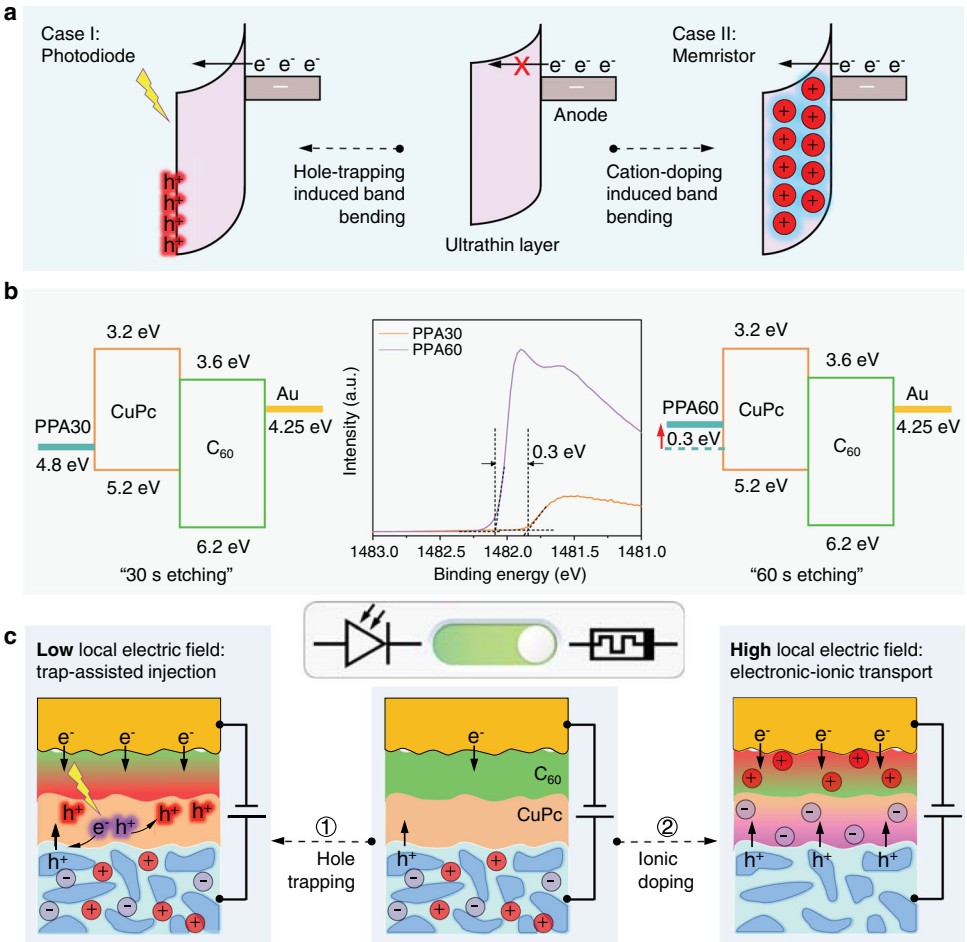

**Fig. 5 Mechanism of the electric-field-driven transition. a** Both trapped holes and accumulated ions in the vicinity of the electrodes have the same effect, i.e., they induce band bending and enhance the charge injection due to the increase in the local carrier density. Red-glowing "h⁺", red circle with "+", and black "e⁻" represent the trapped photogenerated hole, accumulated cation, and injection electron, respectively. **b** Photoelectron spectra (Al Kα-monochromated, 300 W, 1486.6 eV) of the wet-etched PEDPT:PSS:AgNWs and the proposed band structures (before equilibrium) with 30 s-etching and 60 s-etching. From the photoelectron spectra obtained by using monochromatic Al Kα radiation, the calculated values of the work function for PPA30 and PPA60 are 4.8 eV and 4.5 eV, respectively. **c** At a low local electric field across the PPA/CuPc interface, electronic transport dominates, resulting in photomultiplication photodiodes as demonstrated in Fig. 3. At a high local electric field across the PPA/CuPc interface, ion migration is activated, resulting in bipolar memristors as demonstrated in Fig. 4. Therefore, by controlling the PPA/CuPc interface electric field, the molecular device can be switched between the photomultiplication photodiode and bipolar memristor.

the correct secondary electron cut-off. The current-voltage (*I-V*) characteristics were measured at room temperature, using a Keithley 2636 A connected to a probe station, with the tube electrode grounded.

**Optical simulation**. The optical field distribution was calculated by the Transfer Matrix method[26]. The photogenerated exciton distribution in the active layer can be calculated according to the optical field distribution and absorption coefficient of the active layer.

**Reporting summary**. Further information on research design is available in the Nature Research Reporting Summary linked to this article.

## Data availability
The authors declare that all data supporting the findings of this study are available from the corresponding author on request.

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

## Acknowledgements

The authors acknowledge Paul Plocica, Tobias Schürer, Bingkun Sun, Zhe Li, and Yu Hong for their technical support. T.L. and J.Q. acknowledge the support and funding from the China Scholarship Council (CSC). O.G.S. acknowledges the financial support by Leibniz Program of the German Research Foundation (SCHM 1298/26-1). M.K. and M.H. acknowledge the financial support by DFG (KN393/25; KN393/26).

## Author contributions

T.L., F.Z. and O.G.S. conceived the idea. T.L. and F.Z. designed the experiment and analyzed the data. T.L. performed all materials preparation, devices fabrication and measurements. M.H. and M.K performed UPS & XPS measurements. Q.J. performed Raman characterization. V.K.B. contributed to optical measurements. T.L. and F.Z. wrote the manuscript with input from all authors. F.Z. and O.G.S. supervised the work.

## Funding

## Competing interests

The authors declare no competing interests.

**Additional information**

