## [Peer review file · Nature Communications]

REVIEWER COMMENTS

Reviewer #1 (Remarks to the Author):

Zhu and coworkers present a very interesting new field-controlled memristive switching molecular device with integrated microfluidics using conductive polymers, that moves ion-controlled switching closer to wafer scale. It also functions as a photomultiplier. It is clever and novel science, and suitable for publication in Nat Comm following some corrections:

1. Can the authors be confident that silver "leaching" via migration/diffusion of Ag nanoparticles does not contribute to switching in a filamentation type mechanism?
2. The notion of coupled ionic-electronic effects is not very helpful in understanding the mechanism. It would be great if future DFT studies could clarify the mechanism but until then I would urge caution in how the proposed mechanism is presented.
3. On a related point, and my largest concern here, is lack of molecular scale characterisation of the polymer/molecule interface - Raman, microscopy and conductivity measurements look to be very careful and show high quality film but it would improve the usefulness of the m/s to present more atomic scale characterisation, assisted by modelling as necessary, to demonstrate how the interface promotes the ion diffusion mechanism. If not, then control materials should be proposed, both positive and negative, to benchmark the findings for PDOT:PSS as the source of the "right" mobile ions for the switching.

Minor thing:

The paper should be carefully proofread to avoid errors and improve readability - e.g., "inhere a large range of" in the first sentence of the Abstract, the unnecessary "over the years" in the second sentence.

Reviewer #2 (Remarks to the Author):

In this manuscript, the authors report on a integrated molecular diode-like devices that exhibit reconfigurable behavior. More specifically, an ionic reservoir can modify the behavior of the device from a diode to a photodiode. The device behavior is explained in detail. Nevertheless, the manuscript does not present any computational/processing aspect close or beyond the state-of-art. Therefore, I believe that the manuscript is suitable for a more specialized journal.

Reviewer #3 (Remarks to the Author):

The manuscript by Li et al. with the title “Process-Programmable Molecular Junctions Switch between Photomultiplication and Memristive Behaviour in On-Chip Integrated Devices” demonstrate memristive behavior of a new type of molecular junction. They report a new type of molecular electronic device which is very interesting and deserves publication in Nat Commun after the authors have addressed the following points.

- “due to the lack of damage-free micro-nano fabrication techniques, it is still a huge challenge to integrate and control the ionic and electronic functions within the molecular devices in parallel at wafer-scale, which is crucial to develop novel and practical functional molecular devices.” This statement is correct for single-molecule junctions, but not for large area junctions. There are quite a few established fabrication techniques that have withstood the test of time , some are even commercially available (DOI: 10.1088/0953-8984/28/9/094011 ; <https://dx.doi.org/10.1021/jacs.9b12424>; but also the groups of Cahen and Lacroix make highly stable molecular junctions on large scales, group of Chiechi reported impressive ways to fabricate functional molecular devices (including memory) with very high stabilities (see doi: 10.1063/5.0050667 for an overview and citations therein). The introduction should be toned down and include a more balanced discussion

- The authors state “it is reasonable to claim that the ensembles of CuPc/C60 molecules fall into the molecular scale”, but in their explanation of the mechanism of they fall back on semiconductor physics which goes against this statement. Schottky junctions, p-n junctions, band bending, can only form when depletion layers are present (which usually far exceed molecular dimensions), such depletion layers cannot form in molecules. In molecular junctions, HOMO/LUMO levels can shift (energy level alignment) for instance, and since there is only one HOMO/LUMO per molecules, band bending is not possible. The proposed model and explanations suggest that their devices are not truly molecular in nature as ions from the electrode material play a crucial role. The authors should modify their manuscript accordingly

After reading the bold introduction, my expectations were high but ...:

- Lack of statistics. No yields, error bars (reproducibilities) etc are reported. Normally heatmaps, tables with statistics (number of samples, number of devices, number of measurements etc), (log)average curves with (log)standard deviations etc should be all given but not reported here.

- Stability. No endurance, retention, or demonstration of non-volatility are given.
- Fig 2b shows that their technique may be very difficult to scale. I find this not a major issue, but this should be mentioned and compared to other fabrication techniques
- How well do their devices perform with respect to other devices? For instance, in their ref 7 the on/off ratios are higher, but there are more examples of molecular memristors in the literature

In all, I find their devices and some of the concepts very interesting (i.e., role of mobile ions) but the authors should refrain from overhyping, provide proper context and acknowledge relevant prior work, and give a balanced discussion regarding the benefit but also disadvantages of their fabrication method before I can support publication.

Dear Editor Dr. Congcong Huang,

Thank you very much for your efforts and help on our manuscript previously entitled “*Process-Programmable Molecular Junctions Switch between Photomultiplication and Memristive Behaviour in On-Chip Integrated Devices*” (ID: NCOMMS-21-44780). Also, we thank you and all the three reviewers for the valuable suggestions and comments, which are very helpful to improve the quality of our manuscript. We have revised the manuscript and supporting information. The revised parts have been marked in **red font**. Should you have any comments and questions, please don’t hesitate to let us know. Many thanks!

Best regards,

Feng Zhu, Tianming Li and Oliver G. Schmidt

Attached files

1. List of changes in the revised submission..... Page 2-3
2. Response to the comments of Reviewer 1..... Page 4-15
3. Response to the comments of Reviewer 2..... Page 16
4. Response to the comments of Reviewer 3..... Page 17-37

List of Changes:

The changes in the present revised version are listed below:

1. The title of this work is revised to be “On-Chip Integrated Process-Programmable Sub-10 nm Thick Molecular Devices Switching between Photomultiplication and Memristive Behaviour”.
2. On Page 1 in the manuscript (in the *Abstract*), the Abstract is rewritten. Specifically, the statement “However, controlling and manufacturing functional molecular devices are still far from being achievable due to challenges in device construction and integration.” is deleted; and the meaning of molecular devices constructed by ultrathin molecular layers is emphasized.
3. On Page 3 in the manuscript, the *Introduction* part is revised. Specifically, the statement “However, due to the lack of damage-free micro-nano fabrication techniques, it is still a huge challenge to integrate and control the ionic and electronic functions within the molecular devices in parallel at wafer-scale, which is crucial to develop novel and practical functional molecular devices.” is deleted; and a paragraph that describes the achievement in ultrathin molecular layer-based molecular devices is added to highlight the promising potential of molecular devices constructed by ultrathin molecular layers.
4. On Page 4 in the manuscript (in the *Introduction*), a brief discussion on the well-established fabrication techniques for large-area molecular junctions is added.
5. On Page 12 in the manuscript, discussion on determining the resistance switching type is added.
6. On Page 14 in the manuscript, description about the retention and endurance stabilities is added.
7. On Page 14 in the manuscript, a brief discussion on the memristive performance of the molecular memristors in this work is added.
8. On Page 20 in the manuscript, the reference list is updated.
9. On Page 31 in the manuscript, the previous Figure 4h and 4i are replaced by the figures of retention and endurance properties, and the corresponding figure legends are added.
10. On Page S4 in Supplementary Information, a comparison of different techniques for creating sub-10 nm thick molecular devices is presented in Section 1 and Supplementary Table 1.
11. On Page S24 in Supplementary Information, the statistics data is shown in Supplementary Figure 11, and the corresponding description is presented in Section 9.
12. On Page S26 in Supplementary Information, a detailed discussion about the memristive mechanism of the molecular memristors in this work is presented in Section 10, which is based on several solid evidences, i.e., I-V hysteresis sequence (Supplementary Figure 12), switchable diode effect (Supplementary Figure 13), I-V hysteresis shrink during light illumination

(Supplementary Figure 14), source of the mobile ions (Supplementary Figure 15), and transition between photomultiplication and memristive behaviour (Supplementary Figure 16).

13. On Page S35 in Supplementary Information, a comparison of the memristive performance with reported memristors is presented in Section 12 and Supplementary Table 3.
14. On Page S39 in Supplementary Information, the reference list is updated.
15. In the revised manuscript and Supplementary Information, the sequence of all figures and references has been updated.
16. The writing is polished to improve the readability.

Response to the comments of Reviewer #1

Reviewer #1 (Remarks to the Author):

Comments: Zhu and coworkers present a very interesting new field-controlled memristive switching molecular device with integrated microfluidics using conductive polymers, that moves ion-controlled switching closer to wafer scale. It is also functions as photomultiplier. It is clever and novel science, and suitable for publication in Nat Comm following some corrections:

Our response:

Thank you very much for your efforts on our manuscript. We greatly appreciate your advice and comments which are very helpful for improving the quality of this work. We have revised the manuscript according to your suggestions. The revised parts are marked in red. Our response is presented point-to-point in the following.

Question (1.1) Can the authors be confident that silver "leaching" via migration/diffusion of Ag nanoparticles does not contribute to switching in a filamentation type mechanism?

Response (1.1)

We appreciate this comment a lot. Since there is a strong relationship among the Reviewer's first three questions, i.e., all of them focus on the memristive mechanism, we would like to provide a comprehensive reply in our Response (1.1) to clarify the working mechanism, and then make additions to our Response (1.2) and Response (1.3). The phenomena observed in our work, i.e., the I-V hysteresis order (see Figure R1a and d), the linear fitting of double logarithmic I-V plot (see Figure R1b, c, e and f), the switchable diode effect (see Figure R2), and the I-V hysteresis shrink during light illumination (see Figure R3), are usually found in perovskite-based barrier-type memristors, which indicates that hysteretic behaviour of our molecular devices originates from ion polarization-modulated charge injection, rather than the formation/rupture of filaments. On the other hand, we conduct a series of control experiments (i.e., replacing the polymer bottom electrode (see Figure R4a-c), replacing the molecular storage medium (see Figure R4d-f), and washing the etched PEDOT:PSS:Ag NWs electrode over a long time (see Figure R5)), and all the results indicate that (1) the etched PEDOT:PSS:Ag NWs electrode is the only source of ions for resistive switching and (2) the hybrid molecular layer is not the decisive factor for the hysteretic behaviour in our work.

A detailed discussion about the memristive mechanism of our molecular memristor is presented below, which has also been added to the revised Supplementary Information (see Section 10, on Page S26).

Section 10. The memristive mechanism

S10.1 Filamentary vs. barrier modulation

To elucidate the working mechanism, we first compare the I-V hysteresis features of our memristors to those of conventional filamentary memristors, as shown in Figure R1 (also in Supplementary Fig. 12). We fabricated a stacked structure of Ni/SiO₂(1 nm)/Al₂O₃(1 nm)/NiFe, which exhibits a typical filamentation-type hysteresis behaviour^{57, 58} (Figure R1a, also in Supplementary Fig. 12a) owing to the formation and rupture of conductive filaments. In this situation the sequence of the I-V hysteresis follows HRS→LRS→LRS→HRS, where HRS and LRS are the abbreviations of the high resistance state (i.e., OFF state) and low resistance state (i.e., ON state), respectively. Therefore, the memristor can maintain the ON/OFF state when the bias polarity is switched. However, for the molecular devices in our work (as shown in Figure R1, also in Supplementary Fig. 12d), the I-V hysteresis follows the order HRS→LRS→HRS→LRS; i.e., the resistance state changes whenever the bias polarity switches, which is consistent with the behaviour of memristors that rely on a Schottky barrier modulation in perovskite-based devices, such as BiFeO₃ (BFO)⁵⁹, BaTiO (BTO)⁶⁰, CH₃NH₃PbI₃ (MAPbI₃)⁶¹, and so on. For this kind of perovskite memristors, the observed hysteresis is generally accepted to be caused by ion migration-induced barrier modulation, which is supported by experimental results^{60, 61, 62} and numerical simulations⁶³. Under an electric field, mobile ions drift in the perovskite layer and accumulate near the electrode surfaces, leading to p-doping in the perovskite near the anode and n-doping near the cathode. Similarly, a reverse bias can flip the doping configuration by forcing positive and negative ions to drift in the opposite direction. The self-doping effect can influence the electrode/perovskite barrier width and therefore increase or decrease the carrier injection capability, resulting in memristive hysteresis. In perovskite materials, the mobile ions originate from lattice defects at the grain boundaries, for example, positively charged I vacancies and negatively charged Pb and MA vacancies in MAPbI₃⁶¹.

Additional evidence for clarifying the mechanism is provided by fitting the double logarithmic I-V plot with a power law⁶⁴, as shown in Figure R1 (also in Supplementary Fig. 12). For filamentation-type memristors, the ON state is attributed to the formation of conductive filaments between the two electrodes, which causes a “short circuit”. Therefore, the charge transport of the On-state is governed by Ohmic conduction, and the corresponding I-V plots of the LRS for both negative and positive

voltages are characterized by Ohmic behaviour ($I \sim V$)^{65, 66}, i.e., the slope is close to 1 (as shown in Figure R1b and 1c, also in Supplementary Fig. 12b and 12c). However, for the molecular devices in our work (as shown in Figure R1e and 1f, also in Supplementary Fig. 12e and 12f) the slopes of the On-state I-V plots at negative and positive voltages are calculated to be 2.1 and 3.2, respectively, which deviate significantly from Ohmic behaviour. Instead, the linear fitting matches well with perovskite-based memristive switching⁶⁰.

Figure R1 | Two types of bipolar memristors according to the I-V hysteresis sequence. a, I-V hysteresis follows the order of HRS→LRS→LRS→HRS. The example shown here is based on a stacked structure of Ni/SiO₂(1 nm)/Al₂O₃(1 nm)/NiFe. b and c, Double logarithmic fittings of the negative part (b) and positive part (c) of the I-V plot shown in a. d, I-V hysteresis follows the order of HRS→LRS→HRS→LRS. The example shown here is based on our PPA60/CuPc(3 nm)/C₆₀(3 nm)/Au. e and f, Double logarithmic fittings of the negative part (e) and positive part (f) of the I-V plot shown in d.

S10.2 Switchable diode effect

The switchable diode effect that usually appears in interface-type perovskite memristors^{59, 67} is also observed in our molecular devices (as shown in Figure R2, also in Supplementary Fig. 13), which has not yet been reported in filamentation-type memristors as far as we know. For the virgin state, the current is very small and nearly symmetric over a small bias range (Figure R2a, also in

Supplementary Fig. 13a); however, when the sweep starts from a higher negative voltage (-1.2 V here), the current increases rapidly in the negative voltage region but increase slowly in the positive voltage region, indicating a reverse diode behaviour (Figure R2b, also in Supplementary Fig. 13b); whereas, when the sweep starts from a higher positive voltage (+1.2 V here), the current shows a forward diode (Figure R2c, also in Supplementary Fig. 13c). The mechanism of the switchable diode effect in perovskite devices is still under debate: some researchers claim that it originates from the polarization of mobile ions^{61, 62, 68}; while some others claim that it is governed by the polarization of ferroelectric domains^{59, 69}. However, both of them agree that the polarized charges (ions or charged centres) under the applied electric field can modify the electrode/perovskite Schottky barriers, leading to switchable diode behaviour. It should be pointed out that in our case both CuPc and C₆₀ are not ferroelectric materials.

Figure R2 | Voltage-polarity-dependent rectifying behaviour. **a**, Virgin state extracted from Figure 4a1. **b** and **c**, Reverse diode (**b**) and forward diode (**c**) behaviours extracted from Figure 4a2. **d**, Single sweep from PPA60 bottom electrode to Au tubular top electrode, resulting in a diode-like rectifying behaviour with negative voltage as the forward bias. **e**, Single sweep from Au tubular top electrode to PPA30 bottom electrode, resulting in a diode-like rectifying behaviour with positive

voltage as the forward bias. The bottom PPA60 electrode is grounded.

S10.3 Photoresponse of the molecular memristors with I-V hysteresis

As shown in Figure R3 (also in Supplementary Fig. 14), compared to the dark condition, the I-V hysteresis almost disappears under illumination. The shrinkage of the I-V hysteresis under illumination has also been reported in perovskite-based solar cells and photodetectors due to ion-involved phenomena^{69, 70}. Most of the mobile ions drift toward the contacts under a high external electric field, and in turn, the ion accumulation impacts the electron and hole distributions (i.e., ion accumulation-induced doping effect), which is deleterious to the performance of the devices in the photovoltaic and light detection fields. Owing to the presence of a very large electron/hole concentration at the interface (where photogenerated holes/electrons would be extracted), these accumulated ions at the electrode interfaces can act as electron-hole recombination centers (i.e., traps) to enhance the interfacial charge recombination during the collection of photogenerated carriers⁷¹. As a consequence, the barrier modulation effect induced by ion accumulation is weakened, hence, the photocurrent is reduced⁷² and the I-V hysteresis reduces under illumination after polarization. Furthermore, the presence of ions at the interfaces can lead to chemical reactions at the contacts, which could also modify the extraction properties. Back to our case, it is supposed that both the enhanced surface recombination and chemical reaction contribute to the shrinkage of the I-V hysteresis. Because after turning off the light, the I-V hysteresis in the dark is severely reduced, especially for the negative part, which is possibly ascribed to the lack of Ag ions owing to the photoreduction of Ag ions with the help of photogenerated electrons.

Figure R3 | Photoresponse of the molecular device based on PPA60/CuPc (5 nm)/C₆₀ (5 nm)/Au tube. a, I-V characteristics in dark. b, I-V characteristics under illumination. c, I-V characteristics in the dark again.

S10.4 Determining the source of the mobile ions

All the abovementioned phenomena indicate that our memristive switching performance stems from the ion migration-induced barrier modulation rather than the formation/rupture of filaments. Usually, interface-type memristors have been widely reported for perovskite-based materials, in which the mobile ions originate from lattice defects. However, in our work, a series of experiments point out that the mobile ions are provided by the etched PEDOT:PSS:Ag NWs, i.e., they are not from the organic hybrid layer CuPc/C₆₀. As shown in Figure R4a and 4b (also in Supplementary Fig. 15a and 15b), when the etched PEDOT:PSS:Ag NWs electrode is replaced by an ITO or Ag film, the I-V hysteresis is negligible. However, when the Ag film electrode is etched for a short time (~2 s) with the same etching solution, the hysteretic behaviour occurs again, but it is very unstable (Figure R4c, also in Supplementary Fig. 15c). This result directly implies that the etched Ag film can provide ions but cannot control the ions as effectively as etched PEDOT:PSS:Ag NWs during the repeated sweeping. Moreover, when CuPc is replaced by H₂PC, a similar I-V hysteresis is also observed (Figure R4d, also in Supplementary Fig. 15d), which not only indicates that the coordinated metallic ions Cu(II) in CuPc do not contribute to the hysteretic phenomenon but also demonstrates that the etched PEDOT:PSS:Ag NWs electrode indeed matters. We also prepared PPA60-based devices with a single CuPc or C₆₀ layer as the active material (Figure R4e and 4f, also in Supplementary Fig. 15e and 15f), and find that both PPA60/CuPc(3 nm)/Au and PPA60/C₆₀(3 nm)/Au exhibit I-V hysteresis. However, the hysteresis loops are much weaker than that of PPA60/CuPc(3 nm)/C₆₀(3 nm)/Au. This observation emphasizes that the ultrathin CuPc/C₆₀ hybrid layer is very important to achieve excellent resistance switching performance, and also implies that the inserted molecules are not the core factor causing the hysteresis, but instead the etched PEDOT:PSS:Ag NWs electrode.

Figure R4 | Determining the origin of the mobile ions. **a**, I-V characteristics of the molecular device based on ITO/CuPc (3 nm)/C₆₀ (3 nm)/Au tube. **b** and **c**, I-V characteristics of the molecular devices based on Ag (40 nm)/CuPc (3 nm)/C₆₀ (3 nm)/Au tube without etching the Ag film (**b**) and with etching the Ag film for a short time (**c**). **d-f**, I-V characteristics of the molecular devices based on PPA60/H₂Pc (3 nm)/C₆₀ (3 nm)/Au tube (**d**), PPA60/CuPc (3 nm)/Au tube (**e**), and PPA60/C₆₀ (3 nm)/Au tube (**f**), respectively.

S10.5 Transition between photomultiplication photodiodes and bipolar memristors

Another experimental evidence is that when increasing the water washing time (from 10 s to 10 min) of PPA60 after wet etching (i.e., before molecule deposition), the corresponding molecular devices act as photomultiplication photodiodes again, and the hysteresis behaviour is suppressed (Figure R5, also in Supplementary Fig. 16). This is because the longer washing procedure partly removes the mobile ions, leading to a decrease in ion concentration and a reduction of the PPA60/CuPc barrier height, both of which make ion transport difficult. This phenomenon further supports the fact that the etched PEDOT:PSS:AgNWs film is the source of mobile ions. In fact, PEDOT:PSS has been extensively used as an “ion reservoir” to store ions. We use the Raman spectrum and light absorption coefficient profile to characterize the PEDOT:PSS:AgNWs film before and after etching, and find

that after etching there are no metallic Ag related signals (see Figure 2f and g in the manuscript). However, the XPS spectrum exhibits a clear Ag signal (see Figure 2h in the manuscript), and the relative atomic content increases with the etching time. If the Ag signal comes from metallic Ag, then the content should decrease as the etching time is prolonged. Obviously, this did not happen. Based on these results, we conclude that Ag nanowires are completely gone after etching and the Ag ions are adsorbed on the etched PEDOT:PSS. Moreover, N is also detected, and the relative atomic content increases with the etching time, which stems from the etching component HNO₃. The mixed etching solution was H₃PO₄ (3):HNO₃ (3):CH₃COOH (23): H₂O (1). In fact, both Ag₃PO₄ and CH₃COOAg exist in solid forms, which may be the reason why element P is not detected.

Figure R5 | Transition between photomultiplication photodiodes and bipolar memristors. a, I-V characteristics of the molecular device based on PEDOT:PSS:AgNWs film wet-etched for 30 s and washed with water for 10 s. b, I-V characteristics of the molecular device based on PEDOT:PSS:AgNWs film wet-etched for 60 s and washed with water for 10 s. c, I-V characteristics of molecular device based on PEDOT:PSS:AgNWs film wet-etched for 60 s and washed with water for 10 min.

In summary, we have clarified that our memristive switching performance is attributed to ion migration-induced barrier modulation (instead of the formation/rupture of filaments), and we have provided evidence that the mobile ions originate from the etched PEDOT:PSS film (rather than the organic hybrid layer CuPc/C₆₀).

Related references in the Supplementary Information:

57. Lee, B.-H., Bae, H., Seong, H., Lee, D.-I., Park, H., Choi, Y.J., Im, S.-G., Kim, S.O., Choi, Y.-K. Direct observation of a carbon filament in water-resistant organic memory. *ACS Nano* **9**, 7306-7313

(2015).

58. Raeis Hosseini, N., Lee, J.-S. Resistive switching memory based on bioinspired natural solid polymer electrolytes. *ACS Nano* **9**, 419-426 (2015).
59. Hong, S., Choi, T., Jeon, J.H., Kim, Y., Lee, H., Joo, H.Y., Hwang, I., Kim, J.S., Kang, S.O., Kalinin, S.V. Large resistive switching in ferroelectric BiFeO₃ nano-island based switchable diodes. *Adv. Mater.* **25**, 2339-2343 (2013).
60. Chen, A., Zhang, W., Dedon, L.R., Chen, D., Khatkhatay, F., MacManus-Driscoll, J.L., Wang, H., Yarotski, D., Chen, J., Gao, X., Martin, L.W., Roelofs, A., Jia, Q. Couplings of Polarization with Interfacial Deep Trap and Schottky Interface Controlled Ferroelectric Memristive Switching. *Adv. Funct. Mater.* **30**, 2000664 (2020).
61. Xiao, Z., Yuan, Y., Shao, Y., Wang, Q., Dong, Q., Bi, C., Sharma, P., Gruverman, A., Huang, J. Giant switchable photovoltaic effect in organometal trihalide perovskite devices. *Nat. Mater.* **14**, 193-198 (2015).
62. Yang, C.H., Seidel, J., Kim, S.Y., Rossen, P.B., Yu, P., Gajek, M., Chu, Y.H., Martin, L.W., Holcomb, M.B., He, Q., Maksymovych, P., Balke, N., Kalinin, S.V., Baddorf, A.P., Basu, S.R., Scullin, M.L., Ramesh, R. Electric modulation of conduction in multiferroic Ca-doped BiFeO₃ films. *Nat. Mater.* **8**, 485-493 (2009).
63. Sajedi Alvar, M., Blom, P.W.M., Wetzelaer, G.A.H. Space-charge-limited electron and hole currents in hybrid organic-inorganic perovskites. *Nat. Commun.* **11**, 4023 (2020).
64. Lim, E., Ismail, R. Conduction Mechanism of Valence Change Resistive Switching Memory: A Survey. *Electronics* **4**, 586-613 (2015).
65. Liu, L., Li, Y., Huang, X., Chen, J., Yang, Z., Xue, K.H., Xu, M., Chen, H., Zhou, P., Miao, X. Low Power Memristive Logic Device Enabled by Controllable Oxidation of 2D HfSe₂ for In-Memory Computing. *Adv. Sci.* **8**, 2005038 (2021).
66. Li, Y., Zhou, Y.-X., Xu, L., Lu, K., Wang, Z.-R., Duan, N., Jiang, L., Cheng, L., Chang, T.-C., Chang, K.-C. Realization of functional complete stateful Boolean logic in memristive crossbar. *ACS Appl. Mater. Interfaces* **8**, 34559-34567 (2016).
67. Jiang, A.Q., Wang, C., Jin, K.J., Liu, X.B., Scott, J.F., Hwang, C.S., Tang, T.A., Lu, H.B., Yang, G.Z. A resistive memory in semiconducting BiFeO₃ thin-film capacitors. *Adv. Mater.* **23**, 1277-1281 (2011).
68. Lee, J.H., Jeon, J.H., Yoon, C., Lee, S., Kim, Y.S., Oh, T.J., Kim, Y.H., Park, J., Song, T.K., Park, B.H. Intrinsic defect-mediated conduction and resistive switching in multiferroic BiFeO₃ thin films epitaxially grown on SrRuO₃ bottom electrodes. *Appl. Phys. Lett.* **108**, 112902 (2016).
69. Choi, T., Lee, S., Choi, Y.J., Kiryukhin, V., Cheong, S.W. Switchable ferroelectric diode and photovoltaic effect in BiFeO₃. *Science* **324**, 63-66 (2009).
70. Kwon, K.C., Hong, K., Van Le, Q., Lee, S.Y., Choi, J., Kim, K.B., Kim, S.Y., Jang, H.W. Inhibition of ion migration for reliable operation of organolead halide perovskite-based Metal/Semiconductor/Metal broadband photodetectors. *Adv. Funct. Mater.* **26**, 4213-4222 (2016).
71. Lan, D. The physics of ion migration in perovskite solar cells: Insights into hysteresis, device performance, and characterization. *Prog. Photovolt. Res. Appl.* **28**, 533-537 (2020).
72. Lan, C., Zou, H., Wang, L., Zhang, M., Pan, S., Ma, Y., Qiu, Y., Wang, Z.L., Lin, Z. Revealing Electrical Poling-Induced Polarization Potential in Hybrid Perovskite Photodetectors. *Adv. Mater.* **32**, 2005481 (2020).

Accordingly, the following description has been added to the revised manuscript (on Page 12, Line 237).

The resistance switching in the PPA60/CuPc/C₆₀/Au tube structure is further explored. As described in detail in Supplementary Section 10, the I-V hysteresis order and linear fitting (Figure R1, also in Supplementary Fig. 12), the switchable diode effect (Figure R2, also in Supplementary Fig. 13), and the hysteresis reduction during light illumination (Figure R3, also in Supplementary Fig. 14) suggest that the hysteretic behaviour of our molecular devices is analogous to that of barrier modulation-related perovskite memristors, instead of conventional filamentary memristors. For memristors based on perovskite materials, e.g., BiFeO₃ films with defects such as bismuth and oxygen deficiencies (acting as mobile ions)^{34, 35}, the hysteresis is attributed to the Schottky barrier modulation induced by ion polarization in the perovskite storage medium.

Related references in the manuscript:

34. Yang, C.H., Seidel, J., Kim, S.Y., Rossen, P.B., Yu, P., Gajek, M., Chu, Y.H., Martin, L.W., Holcomb, M.B., He, Q., Maksymovych, P., Balke, N., Kalinin, S.V., Baddorf, A.P., Basu, S.R., Scullin, M.L., Ramesh, R. Electric modulation of conduction in multiferroic Ca-doped BiFeO₃ films. *Nat. Mater.* **8**, 485-493 (2009).
35. Xiao, Z., Yuan, Y., Shao, Y., Wang, Q., Dong, Q., Bi, C., Sharma, P., Gruverman, A., Huang, J. Giant switchable photovoltaic effect in organometal trihalide perovskite devices. *Nat. Mater.* **14**, 193-198 (2015).

Question (1.2) *The notion of coupled ionic-electronic effects is not very helpful in understanding the mechanism. It would be great if future DFT studies could clarify the mechanism but until then I would urge caution in how the proposed mechanism is presented.*

Response (1.2)

We greatly appreciate your advice and concern. As mentioned in our **Response (1.1)** to your first **Question**, the mechanism of the hysteretic behaviour in our molecular devices is proposed to be the ion accumulation-enhanced charge injection. One interesting point is, in fact, that the HRS→LRS→HRS→LRS hysteresis sequence, the switchable diode effect, and the hysteresis reduction upon light illumination are usually found in perovskite-based memristors/solar cells, which are attributed to the Schottky barrier modulation induced by the ion polarization in the perovskites. All of these phenomena occur in our molecular devices; therefore, we rely on existing phenomena in the field of perovskite memristors to explain the working mechanisms in our devices.

Question (1.3) *On a related point, and my largest concern here, is lack of molecular scale characterisation of the polymer/molecule interface - Raman, microscopy and conductivity*

measurements look to be very careful and show high quality film but it would improve the usefulness of the m/s to present more atomic scale characterisation, assisted by modelling as necessary, to demonstrate how the interface promotes the ion diffusion mechanism. If not, then control materials should be proposed, both positive and negative, to benchmark the findings for PEDOT:PSS as the source of the "right" mobile ions for the switching.

Response (1.3)

We have taken your advice and carried out a series of control experiments to clarify the working mechanisms. As illustrated in the Response (1.1) to your first **Question**, we replaced the etched PEDOT:PSS:Ag NWs bottom electrode with inorganic material, replaced the CuPC/C₆₀ hybrid layer with other components, and washed the etched PEDOT:PSS:Ag NWs for extended periods. According to the experimental results, we conclude that (1) the CuPC/C₆₀ hybrid layer is not the origin of the hysteretic behaviour, and (2) the etched PEDOT:PSS:Ag NWs electrode leads to the hysteresis phenomenon.

Alvar *et al.*, [Nat. Commun. 11, 4023 (2020)] used the electronic-ionic drift-diffusion model (via MATLAB) to simulate the hysteresis in the archetypical perovskite methyl ammonium lead iodide (MAPbI₃), and revealed that the coupling between ion migration and electron/hole injection gave rise to the hysteresis in the order of HRS→LRS→HRS→LRS (as shown in Figure R6). Although the system described by their model is much simpler than ours, it provides a valuable reference and supports the interpretation in our work.

Figure R6 | Current density-voltage characteristics of a hole-only device at 275 K. The current density-voltage characteristics (solid lines) are simulated with the same set of parameters under two different conditions for ionic charges: mobile positive ions and a uniform density of immobile negative ions (a) and mobile negative ions and immobile positive ions (b). Source: Sajedi Alvar M,

Blom PWM, Wetzelaer GAH. Space-charge-limited electron and hole currents in hybrid organic-inorganic perovskites. Nat Commun 11, 4023 (2020).

Question (1.4) *Minor thing:*

The paper should be carefully proofread to avoid errors and improve readability - e.g., "inhere a large range of" in the first sentence of the Abstract, the unnecessary "over the years" in the second sentence.

Response (1.4)

Thank you very much for your suggestion. In the revised manuscript, we carefully polished our language and hope the manuscript is more readable now.

Response to the comments of Reviewer #2

Reviewer #2 (Remarks to the Author):

Comments: In this manuscript, the authors report on a integrated molecular diode-like devices that exhibit reconfigurable behaviour. More specifically, an ionic reservoir can modify the behaviour of the device from a diode to a photodiode. The device behaviour is explained in detail. Nevertheless, the manuscript does not present any computational/processing aspect close or beyond the state-of-art. Therefore, I believe that the manuscript is suitable for a more specialized journal.

Our response:

We would like to thank Reviewer #2 for reading our manuscript. However, we do not “modify the behaviour of the device from a diode to a photodiode”. Instead, we apply rolled-up nanotechnology to develop and integrate new, i.e. process-programmable functional molecular devices on wafer-scale, which can switch between photomultiplication photodiodes and bipolar memristors. The transition depends on the release of mobile ions stored in the bottom polymeric electrode, and can be controlled by modulating the local electric field at the interface between the ultrathin molecular layer and the bottom polymer electrode. From the aspect of processing, it is the first time to realize fully integrated function-switching molecular devices at the sub-10 nm thickness scale by coupling electronic and ionic transport mechanisms. We believe this work goes way beyond state-of-the-art in small-scale organic devices, and we hope it can attract broad interest in the fields of molecular electronics, organic electronics, materials sciences, nanotechnologies and electronics engineering.

Response to the comments of Reviewer #3

Reviewer #3 (Remarks to the Author):

Comments: The manuscript by Li et al. with the title “Process-Programmable Molecular Junctions Switch between Photomultiplication and Memristive Behaviour in On-Chip Integrated Devices” demonstrate memristive behaviour of a new type of molecular junction. They report a new type of molecular electronic device which is very interesting and deserves publication in Nat Commun after the authors have addressed the following points.

Our response:

Thank you very much for your efforts on our manuscript. We greatly appreciate your advice and comments which are very helpful for improving the quality of this work. We have revised the manuscript according to your suggestions. The response and revised parts are marked in blue and red, respectively. Our response is presented point-to-point in the following.

Question (3.1) *“due to the lack of damage-free micro-nano fabrication techniques, it is still a huge challenge to integrate and control the ionic and electronic functions within the molecular devices in parallel at wafer-scale, which is crucial to develop novel and practical functional molecular devices.” This statement is correct for single-molecule junctions, but not for large area junctions. There are quite a few established fabrication techniques that have withstood the test of time , some are even commercially available (DOI: 10.1088/0953-8984/28/9/094011 ; <https://dx.doi.org/10.1021/jacs.9b12424>; but also the groups of Cahen and Lacroix make highly stable molecular junctions on large scales, group of Chiechi reported impressive ways to fabricate functional molecular devices (including memory) with very high stabilities (see doi: 10.1063/5.0050667 for an overview and citations therein). The introduction should be toned down and include a more balanced discussion.*

Response (3.1)

We are very grateful for your suggestion. We agree that it is not proper to skip the well-established fabrication techniques in the introduction. Accordingly, we modified our introduction part and now provide a more balanced discussion. Moreover, we prepared a table (in Supplementary Section 1) to compare the different fabrication techniques.

1) In the **Abstract**, we deleted the sentence “However, controlling and manufacturing functional

molecular devices are still far from being achievable due to challenges in device construction and integration.”

- 2) In the *Introduction*, we deleted the sentence “However, due to the lack of damage-free micro-nano fabrication techniques, it is still a huge challenge to integrate and control the ionic and electronic functions within the molecular devices in parallel at wafer-scale, which is crucial to develop novel and practical functional molecular devices.”
- 3) In the *Introduction* (on Page 4, Line 69), we provided a more balanced discussion about different fabrication techniques, as follows:

Molecular junctions are the most investigated sub-10 nm thick molecular devices. Compared to single-molecule junctions, large-area junctions based on molecular ensembles are more promising for scalable fabrication. Various fascinating techniques have been developed to realize ensemble molecular junctions on large scales, such as polydimethylsiloxane (PDMS)-assisted liquid metal contacts¹⁴, suspended nanowire junctions¹⁵, surface-diffusion-mediated deposition¹⁶, nanoskiving¹⁷, carbon paint protective layer-based junctions¹⁸, and electron-beam deposition of carbon¹⁹. A comparison of these techniques for large-area sub-10 nm thick molecular devices is discussed in Supplementary Section 1 and Supplementary Table 1. Among them, rolled-up nanotechnology provides an efficient strategy to fabricate fully integrated functional molecular devices on chip via damage-free soft contacts²⁰, which is not only suitable for ordered self-assembled monolayers but also for deposited ultrathin molecular layers. The latter allows for great flexibility in designing the molecular component and thickness through standard vacuum deposition technologies such as thermal evaporation^{13, 21}.

Related references in the manuscript:

13. Li, T., Bandari, V.K., Hantusch, M., Xin, J., Kuhrt, R., Ravishankar, R., Xu, L., Zhang, J., Knupfer, M., Zhu, F., Yan, D., Schmidt, O.G. Integrated molecular diode as 10 MHz half-wave rectifier based on an organic nanostructure heterojunction. *Nat. Commun.* **11**, 3592 (2020).
14. Karuppanan, S.K., Hongting, H., Troadec, C., Vilan, A., Nijhuis, C.A. Ultrasoft and Photoresist-Free Micropore-Based EGaIn Molecular Junctions: Fabrication and How Roughness Determines Voltage Response. *Adv. Funct. Mater.* **29**, 1904452 (2019).
15. Kayser, B., Fereiro, J.A., Bhattacharyya, R., Cohen, S.R., Vilan, A., Pecht, I., Sheves, M., Cahen, D. Solid-state electron transport via the protein azurin is temperature-independent down to 4 K. *J. Phys. Chem. Lett.* **11**, 144-151 (2019).
16. Bonifas, A.P., McCreery, R.L. ‘Soft’Au, Pt and Cu contacts for molecular junctions through surface-diffusion-mediated deposition. *Nat. Nanotechnol.* **5**, 612-617 (2010).
17. Pourhossein, P., Chiechi, R.C. Directly addressable sub-3 nm gold nanogaps fabricated by nanoskiving using self-assembled monolayers as templates. *ACS Nano* **6**, 5566-5573 (2012).

18. Karuppannan, S.K., Neoh, E.H.L., Vilan, A., Nijhuis, C.A. Protective layers based on carbon paint to yield high-quality large-area molecular junctions with low contact resistance. *J. Am. Chem. Soc.* **142**, 3513-3524 (2020).
19. Bergren, A.J., Zeer-Wanklyn, L., Semple, M., Pekas, N., Szeto, B., McCreery, R.L. Musical molecules: the molecular junction as an active component in audio distortion circuits. *J. Phys. Condens. Matter* **28**, 094011 (2016).
20. Bufon, C.C., Gonzalez, J.D., Thurmer, D.J., Grimm, D., Bauer, M., Schmidt, O.G. Self-assembled ultra-compact energy storage elements based on hybrid nanomembranes. *Nano Lett.* **10**, 2506-2510 (2010).
21. Jalil, A.R., Chang, H., Bandari, V.K., Robaschik, P., Zhang, J., Siles, P.F., Li, G., Burger, D., Grimm, D., Liu, X., Salvan, G., Zahn, D.R., Zhu, F., Wang, H., Yan, D., Schmidt, O.G. Fully Integrated Organic Nanocrystal Diode as High Performance Room Temperature NO₂ Sensor. *Adv. Mater.* **28**, 2971-2977 (2016).

4) In *Supplementary Information* (on Page S4), we provide a table to compare different fabrication techniques (see Section 1 and Supplementary Table 1), as follows:

Section 1. Comparison of different techniques for creating sub-10 nm thick molecular devices

A literature study is conducted to compare the rolled-up soft contact technology with other fabrication techniques for creating sub-10 nm thick molecular devices, as shown in Supplementary Table 1. Each technique contributes greatly to the development of molecular electronics^{20, 21}.

Bending induced by releasing built-in strain gradients in thin film layer stacks is an effective approach to self-assemble 2D planar nanomembranes into 3D micro- and nano-tubular architectures, which has provided a smart platform for various fundamental investigations as well as promising applications in the field of the micro- and nanosciences, including magnetics²², biology²³, optics²⁴, robotics²⁵, and energy storage²⁶. At the same time, this self-curling effect can also create a damage-free and self-adjusted top electrode on molecular layers (including self-assembled monolayers, deposited ultrathin molecular layers, and so on) after rolling, hence extending this technique to molecular electronics²⁷. More importantly, the self-rolling process is compatible to conventional photolithography techniques. Therefore, fully integrated sub-10 nm thick molecular devices on wafer scale can be realized by rolled-up soft contacts. Based on this technique some significant achievements have been obtained.

The capability of rolled-up soft contacts is comparable to existing well-established fabrication techniques for creating sub-10 nm thick molecular devices on large scales such as nanoskiving²⁸, surface-diffusion-mediated deposition²⁹, pore-assisted carbon paint³⁰, and so on. Still, there are

challenges to tackle when using the rolled-up nanotechnology. First, due to the ultrathin feature of the sacrificial and strained layers, any defect can lead to an uneven stress distribution, resulting in the failure of rolling. Second, the roughness of the top tubular and bottom finger electrode is usually in the same range of the molecular film thickness. Therefore, the thinner the molecular layer, the more likely the device becomes shorted. Consequently, to obtain a high yield of working devices, much attention should be paid to carefully control the quality and roughness of every film.

Supplementary Table 1 | Comparison of different techniques for creating sub-10 nm thick molecular devices

Techniques	Key principle	Efficiency ^a	Structure stability	Addressability ^b	Fabrication scalability	Fabrication difficulty	Ref.
Conductive AFM	Metal-coated AFM tip	Low	Low	Difficult	No	Low	31
Crossed wire junction	Lorentz force	Low	Low	Easy	No	Low	32
Liquid metal contact	Surface tension	Low to medium	Low	Difficult	Possible (PDMS channel)	Medium	33, 34
Suspended nanowire junctions	Dielectrophoretic trapping	Medium	High	Difficult	Possible (randomly distributed)	Medium	35, 36
E-beam deposited carbon film	E-beam deposition	Low to medium	High	Easy	High	Low	37, 38
Surface-diffusion-mediated deposition	Atom diffusion	High	High	Difficult	High	High	29
Nanoskiving	Edge lithography	High	High	Easy	High	Medium to high	39
Pore + metal evaporation	Photolithography, metal evaporation	Low	High	Easy	High	High	40
Pore + conductive carbon/polymer	Photolithography, spin coating	High	High	Easy	High	High	30, 41
Lift-and-float approach	Liftoff and floating	Medium	High	Difficult	Possible (polymer-assisted)	Low	42

Metal transfer printing	PDMS Stamp	Medium to high	High	Easy	High (relying on the stamp)	Medium	43
Graphene transfer	Adhesive tape	Medium	High	Easy	Possible	Medium	44
Rolled-up soft contact	Photolithography, stress release	Medium to high	High	Easy	High (full integration)	High	This work

a, Efficiency: Sum of massive fabrication capability and yield.

b, Addressability: Sum of the capability to settle the molecules, to locate the top electrodes, and to measure the molecular junctions.

Related references in the Supplementary Information:

20. Haick, H., Cahen, D. Making contact: Connecting molecules electrically to the macroscopic world. *Prog. Surf. Sci.* **83**, 217-261 (2008).
21. Liu, Y., Qiu, X., Soni, S., Chiechi, R.C. Charge transport through molecular ensembles: Recent progress in molecular electronics. *Chem. Phys. Rev.* **2**, 021303 (2021).
22. Gabler, F., Karnaushenko, D.D., Karnaushenko, D., Schmidt, O.G. Magnetic origami creates high performance micro devices. *Nat. Commun.* **10**, 1-10 (2019).
23. Akbar, F., Rivkin, B., Aziz, A., Becker, C., Karnaushenko, D.D., Medina-Sánchez, M., Karnaushenko, D., Schmidt, O.G. Self-sufficient self-oscillating microsystem driven by low power at low Reynolds numbers. *Sci. Adv.* **7**, eabj0767 (2021).
24. Yin, Y., Wang, J., Wang, X., Li, S., Jorgensen, M.R., Ren, J., Meng, S., Ma, L., Schmidt, O.G. Water nanostructure formation on oxide probed in situ by optical resonances. *Sci. Adv.* **5**, eaax6973 (2019).
25. Bandari, V.K., Nan, Y., Karnaushenko, D., Hong, Y., Sun, B., Striggow, F., Karnaushenko, D.D., Becker, C., Faghieh, M., Medina-Sánchez, M. A flexible microsystem capable of controlled motion and actuation by wireless power transfer. *Nat. Electron.* **3**, 172-180

(2020).

26. Lee, Y., Bandari, V.K., Li, Z., Medina-Sánchez, M., Maitz, M.F., Karnaushenko, D., Tsurkan, M.V., Karnaushenko, D.D., Schmidt, O.G. Nano-biosupercapacitors enable autarkic sensor operation in blood. *Nat. Commun.* **12**, 1-10 (2021).
27. Li, T., Bandari, V.K., Hantusch, M., Xin, J., Kuhrt, R., Ravishankar, R., Xu, L., Zhang, J., Knupfer, M., Zhu, F., Yan, D., Schmidt, O.G. Integrated molecular diode as 10 MHz half-wave rectifier based on an organic nanostructure heterojunction. *Nat. Commun.* **11**, 3592 (2020).
28. Pourhossein, P., Chiechi, R.C. Directly addressable sub-3 nm gold nanogaps fabricated by nanoskiving using self-assembled monolayers as templates. *ACS Nano* **6**, 5566-5573 (2012).
29. Bonifas, A.P., McCreery, R.L. 'Soft' Au, Pt and Cu contacts for molecular junctions through surface-diffusion-mediated deposition. *Nat. Nanotechnol.* **5**, 612-617 (2010).
30. Karuppanan, S.K., Neoh, E.H.L., Vilan, A., Nijhuis, C.A. Protective layers based on carbon paint to yield high-quality large-area molecular junctions with low contact resistance. *J. Am. Chem. Soc.* **142**, 3513-3524 (2020).
31. Hnid, I., Frath, D., Lafalet, F., Sun, X., Lacroix, J.-C. Highly efficient photoswitch in diarylethene-based molecular junctions. *J. Am. Chem. Soc.* **142**, 7732-7736 (2020).
32. Seferos, D.S., Trammell, S.A., Bazan, G.C., Kushmerick, J.G. Probing \$\pi\$ -coupling in molecular junctions. *Proc. Natl Acad. Sci. USA* **102**, 8821-8825 (2005).
33. Nijhuis, C.A., Reus, W.F., Barber, J.R., Dickey, M.D., Whitesides, G.M. Charge transport and rectification in arrays of SAM-based tunneling junctions. *Nano Lett.* **10**, 3611-3619 (2010).
34. Chiechi, R.C., Weiss, E.A., Dickey, M.D., Whitesides, G.M. Eutectic gallium–indium (EGaIn): a moldable liquid metal for electrical characterization of self-assembled monolayers. *Angew. Chem. Int. Ed.* **47**, 142-144 (2008).
35. Kayser, B., Fereiro, J.A., Bhattacharyya, R., Cohen, S.R., Vilan, A., Pecht, I., Sheves, M., Cahen, D. Solid-state electron transport via

- the protein azurin is temperature-independent down to 4 K. *J. Phys. Chem. Lett.* **11**, 144-151 (2019).
36. Yoon, H.P., Maitani, M.M., Cabarcos, O.M., Cai, L., Mayer, T.S., Allara, D.L. Crossed-nanowire molecular junctions: a new multispectroscopy platform for conduction--structure correlations. *Nano Lett.* **10**, 2897-2902 (2010).
37. Tefashe, U.M., Nguyen, Q.V., Lafalet, F., Lacroix, J.-C., McCreery, R.L. Robust Bipolar Light Emission and Charge Transport in Symmetric Molecular Junctions. *J. Am. Chem. Soc.* **139**, 7436-7439 (2017).
38. Bergren, A.J., Zeer-Wanklyn, L., Semple, M., Pekas, N., Szeto, B., McCreery, R.L. Musical molecules: the molecular junction as an active component in audio distortion circuits. *J. Phys. Condens. Matter* **28**, 094011 (2016).
39. Pourhossein, P., Vijayaraghavan, R.K., Meskers, S.C., Chiechi, R.C. Optical modulation of nano-gap tunnelling junctions comprising self-assembled monolayers of hemicyanine dyes. *Nat. Commun.* **7**, 1-9 (2016).
40. Wang, W., Lee, T., Reed, M.A. Mechanism of electron conduction in self-assembled alkanethiol monolayer devices. *Phys. Rev. B* **68**, 035416 (2003).
41. Akkerman, H.B., Blom, P.W., de Leeuw, D.M., de Boer, B. Towards molecular electronics with large-area molecular junctions. *Nature* **441**, 69-72 (2006).
42. Vilan, A., Cahen, D. Soft contact deposition onto molecularly modified GaAs. Thin metal film flotation: principles and electrical effects. *Adv. Funct. Mater.* **12**, 795-807 (2002).
43. Loo, Y.-L., Lang, D.V., Rogers, J.A., Hsu, J.W. Electrical contacts to molecular layers by nanotransfer printing. *Nano Lett.* **3**, 913-917 (2003).
44. Wang, Z., Dong, H., Li, T., Hviid, R., Zou, Y., Wei, Z., Fu, X., Wang, E., Zhen, Y., Nørgaard, K. Role of redox centre in charge transport investigated by novel self-assembled conjugated polymer molecular junctions. *Nat. Commun.* **6**, 1-10 (2015).

Question (3.2) *The authors state "it is reasonable to claim that the ensembles of CuPc/C₆₀ molecules fall into the molecular scale", but in their explanation of the mechanism of they fall back on semiconductor physics which goes against this statement. Schottky junctions, p-n junctions, band bending, can only form when depletion layers are present (which usually far exceed molecular dimensions), such depletion layers cannot form in molecules. In molecular junctions, HOMO/LUMO levels can shift (energy level alignment) for instance, and since there is only one HOMO/LUMO per molecules, band bending is not possible. The proposed model and explanations suggest that their devices are not truly molecular in nature as ions from the electrode material play a crucial role. The authors should modify their manuscript accordingly*

Response (3.2)

Thank you very much for this valuable comment, and we appreciate it a lot. Our integrated molecular devices are comparable in size to SAM-based junctions (i.e., below 10 nm), however, as pointed out by Reviewer #3 the working principle of our molecular devices falls into the scope of semiconductor physics. Therefore, in order to better represent and describe the dimension and the mechanism, we now term our devices as "sub-10 nm thick molecular devices", in which the classic molecular junctions are also included. We modified our manuscript accordingly, and the main revised points are as follows:

- 1) We changed our **Title** to "On-Chip Integrated Process-Programmable Sub-10 nm Thick Molecular Devices Switching between Photomultiplication and Memristive Behaviour".
- 2) In the **Abstract** (on Page 1, Line 16), we deleted the sentence "However, controlling and manufacturing functional molecular devices are still far from being achievable due to challenges in device construction and integration.", and we rewrote it as follows:
Molecular junctions based on single molecules or self-assembled monolayers are promising candidates for a new generation of nanoelectronic applications as they not only have the potential to transcend Moore's law but also inhere a large range of both physical and chemical functionalities. However, molecular devices constructed by ultrathin molecular layers with dimensions comparable to ensemble molecular junctions, which can offer alternative approaches to realize functional molecular devices, are often overlooked.
- 3) In the **Introduction** (on Page 3, Line 51), we added a description of the meaning of molecular devices constructed by depositing ultrathin molecular layers with dimensions comparable to ensemble molecular junctions, as follows:

Generally, molecular devices focus on molecular junctions, in which the functionalities are realized by manipulating the chemical properties and electronic structures of the individual or ensemble molecules^{8, 9}. However, the potential of molecular devices constructed by ultrathin molecular layers with dimensions comparable to ensemble molecular junctions has been underestimated. For instance, fundamental studies predicted that short-channel molecular rectifiers can achieve rectification frequencies in the terahertz range^{10, 11}, but a state-of-the-art half-wave SAM rectifier (consisting of Ag^{TS}-SC₁₁Fc₂//Ga₂O₃/EGaIn) was reported to operate at only 50 Hz¹². In marked contrast, Li *et al.* constructed integrated rectifiers based on nanometer-thick hybrid layers consisting of fluorinated cobalt phthalocyanine (F₁₆CoPc)/copper phthalocyanine (CuPc) heterojunctions, which were able to convert alternating current to direct current with a frequency of up to 10 MHz¹³. The excellent high-frequency response was mainly attributed to the band bending of the ultrathin heterojunction. The same dimension but different working principle attracts our special interest to further integrate and control the ionic and electronic functions within ultrathin molecular layer-based devices on the wafer-scale, which is crucial to develop novel and practical functional molecular devices.

Related references in the manuscript:

8. Xin, N., Guan, J., Zhou, C., Chen, X., Gu, C., Li, Y., Ratner, M.A., Nitzan, A., Stoddart, J.F., Guo, X. Concepts in the design and engineering of single-molecule electronic devices. *Nat. Rev. Phys.* **1**, 211-230 (2019).
9. Chen, X., Roemer, M., Yuan, L., Du, W., Thompson, D., Del Barco, E., Nijhuis, C.A. Molecular diodes with rectification ratios exceeding 10⁵ driven by electrostatic interactions. *Nat. Nanotechnol.* **12**, 797-803 (2017).
10. Steudel, S., Myny, K., Arkhipov, V., Deibel, C., De Vusser, S., Genoe, J., Heremans, P. 50 MHz rectifier based on an organic diode. *Nat. Mater.* **4**, 597-600 (2005).
11. Celestin, M., Krishnan, S., Bhansali, S., Stefanakos, E., Goswami, D.Y. A review of self-assembled monolayers as potential terahertz frequency tunnel diodes. *Nano Res.* **7**, 589-625 (2014).
12. Nijhuis, C.A., Reus, W.F., Siegel, A.C., Whitesides, G.M. A molecular half-wave rectifier. *J. Am. Chem. Soc.* **133**, 15397-15411 (2011).
13. Li, T., Bandari, V.K., Hantusch, M., Xin, J., Kuhrt, R., Ravishankar, R., Xu, L., Zhang, J., Knupfer, M., Zhu, F., Yan, D., Schmidt, O.G. Integrated molecular diode as 10 MHz half-wave rectifier based on an organic nanostructure heterojunction. *Nat. Commun.* **11**, 3592 (2020).

- 4) We revised the statement “Compared to the size of a single phthalocyanine molecule (~1.5 nm) and a fullerene molecule (~0.7 nm), it is reasonable to claim that the ensembles of CuPc/C₆₀ molecules fall into the molecular scale” to “It is reasonable to claim that the ensembles of

CuPc/C₆₀ molecules fall into the sub-10 nm thickness scale” (on Page 6, Line 114).

Question (3.3) *After reading the bold introduction, my expectations were high but ...:*

Response (3.3)

Thank you very much for this comment, and we hope our revision and reply can now meet all your expectations.

Question (3.3.1) *Lack of statistics. No yields, error bars (reproducibilities) etc are reported. Normally heatmaps, tables with statistics (number of samples, number of devices, number of measurements etc), (log)average curves with (log)standard deviations etc should be all given but not reported here.*

Response (3.3.1)

Thank you very much for this suggestion. In the revised version, we provide statistical data of the device yield and performance. We find that the yield increases with the thickness of the molecular hybrid layer, and the values of the molecular devices based on CuPc(3 nm)/C₆₀(3 nm), CuPc(5 nm)/C₆₀(5 nm), and CuPc(20 nm)/C₆₀(20 nm) are 39.7%, 48.5%, and 63.2%, respectively. This is because the roughness (~ 2nm) of the top tubular and bottom finger electrodes is on the same order of the molecular film thickness. Therefore, the thinner the molecular layer, the more likely the device becomes shorted, as shown in Figure R7. On the other hand, our yield is greatly limited by the simple laboratory-grade thermal deposition machine, which brings contamination to the deposited metal films, hence increasing the roughness and causing failure in rolling. The latter can lead to ill-contact with the molecular layer and then open the circuit. Consequently, it is feasible to achieve a much higher yield if we can improve the experimental conditions, such as using industrial-grade equipment.

Figure R7 | Details of the devices. a, Rolled-up soft-contact illustrated by AFM images. b and c, AFM topography of rolled-up Au tube (measured on the top surface of the tube). The roughness of the Au tube is extracted from the selected area (in the blue dashed box) along the lateral direction of the tube. d, Schematic illustration of the local contacts. The effective gap between the two electrodes shrinks due to the roughness of the Au finger and the Au tube, as indicated by the blue dashed circle. Adapted from our previous work: *Li, T. et al. Integrated molecular diode as 10 MHz half-wave rectifier based on an organic nanostructure heterojunction. Nat. Commun. 11, 3592 (2020).*

The following description has been added in the revised manuscript (on Page 12, Line 235).

Statistics on yield and resistance switching performance are provided in Supplementary Section 9.

The statistics figure is put as Supplementary Figure 11, and the corresponding description has been added in the revised Supplementary Information (on Page S24), as follows:

Section 9. Statistics

Statistics of the molecular devices are shown in Figure R8 (also in Supplementary Fig. 11). As extracted from Figure R8a-c (also in Supplementary Fig. 11a-c), the yields of the molecular devices based on CuPc(3 nm)/C₆₀(3 nm), CuPc(5 nm)/C₆₀(5 nm), and CuPc(20 nm)/C₆₀(20 nm) are 39.7% (=27/68), 48.5% (=33/68), and 63.2% (=43/68), respectively. This trend in yield versus thickness originates from the fact that the thinner the molecular layer, the more likely the device becomes shorted, because the roughness (~ 2nm) of the top tubular and bottom finger electrodes is in the same

range of the molecular film thickness, which can shrink the effective gap between the two electrodes. Statistics of the device performance are also provided in terms of the rectification ratio, ON/OFF ratio, ON-state current, and switching voltage, as shown in Figure R8d-g (also in Supplementary Fig. 11d-g). Since the hysteretic behaviour of PPA60/CuPc(20 nm)/C₆₀(20 nm)/Au is not remarkable, only the parameters of molecular devices based on CuPc(3 nm)/C₆₀(3 nm) and CuPc(5 nm)/C₆₀(5 nm) are presented.

Figure R8 | Statistics. a-c, Distribution maps of the molecular devices based on CuPc(3 nm)/C₆₀(3 nm), CuPc(5 nm)/C₆₀(5 nm), and CuPc(20 nm)/C₆₀(20 nm), respectively. The green and gray squares represent successful and failed devices on the sample substrates, where a “successful device” refers to a device that exhibits I-V hysteresis behaviour and a “failed device” refers to a shorted or open device. d-g, Average values of the rectification ratio at ±0.5 V (d), ON/OFF ratio at ±0.5 V (e), ON-state current at ±0.5 V (f), and the switching voltage (g). Error bars indicate the standard deviation for each parameter of the working devices. “3/3” and “5/5” denote the molecular devices based on CuPc(3 nm)/C₆₀(3 nm) and CuPc(5 nm)/C₆₀(5 nm), respectively. All data are obtained with the bottom finger electrodes grounded. “Forward” and “Reverse” in d indicate the forward diode and

reverse diode, respectively. “Forward” and “Reverse” in e-g indicate the positive and negative voltage regions, respectively.

Question (3.3.2) Stability. No endurance, retention, or demonstration of non-volatility are given.

Response (3.3.2)

Thank you very much for this comment, and we appreciate it a lot. In the revised manuscript, Figure 4h and 4i in the manuscript are updated with the endurance and retention performances (on Page 31), and the corresponding description has been added (on Page 14, Line 290), as follows:

The stability of our molecular memristors was characterized. The retention time of the PPA60/CuPc(3 nm)/C₆₀(3 nm)/Au device is stable and reliable over 2000 s, and the ON/OFF current ratio remains at ~10³ with a continuous reading bias of 0.5 V, indicating good retention stability (Figure R9h, also in Figure 4h in the manuscript). The endurance of the molecular device degrades slightly over the first 100 sweep cycles due to the decrease in the ON-state current, however, the ON/OFF ratio remains in the range of 10² (Figure R9i, also in Figure 4i in the manuscript).

Figure R9 | Molecular bipolar volatile memristors based on PPA60/CuPc (x nm)/C₆₀ (x nm)/Au tube. h, Retention of ON and OFF states of PPA60/CuPc(3 nm)/C₆₀(3 nm)/Au under

continuous read-out voltages (at 0.5 V). The device is switched to the ON state by applying a positive voltage pulse at 1.5 V for a few seconds. i, Endurance of ON and OFF states of PPA60/CuPc(3 nm)/C₆₀(3 nm)/Au over the first 100 measurement cycles (read at 0.5 V).

Question (3.3.3) *Fig 2b shows that their technique may be very difficult to scale. I find this not a major issue, but this should be mentioned and compared to other fabrication techniques*

Response (3.3.3)

Thank you very much for this comment. Our fabrication is based on multiple photolithography-deposition-lift-off (or etching) steps, which is the same process used in microfabrication for integration. It is possible to increase to device density by optimizing the circuit design, such as decreasing the size and number of the contact pads for external measurements or arranging the contact pads only around the wafer edges. We admit that although our rolled-up tubes provide smart soft-contacts to protect the ultrathin molecular layers on wafer-scale, their micron-scale dimensions (especially the length and width of the microtubes) limit miniaturization to the nanoscale.

As presented in our **Response (3.1)** to your first **Question**, we modified our introduction part and provided a more balanced discussion. And we also prepared a table to compare different fabrication techniques in Supplementary Section 1 and Supplementary Table 1.

Question (3.3.4) *How well do their devices perform with respect to other devices? For instance, in their ref 7 the on/off ratios are higher, but there are more examples of molecular memristors in the literature*

Response (3.3.4)

Thank you very much for this comment, and we appreciate it a lot. To compare our device performance with respect to other devices, we created a table in Supplementary Section 12 (Supplementary Table 3). There are different ways of classifying memristors, such as according to the material properties, the working mechanisms, the fabrication method, and so on. Here, we would like to focus on bipolar memristors and simply divide them into two categories according to the sequence of the I-V hysteresis loops. For your reading convenience, we put the corresponding Figure R10 below, which is a part of the Supplementary Figure 12 in the revised Supplementary Information file.

Figure R10 | Two types of bipolar memristors according to the I-V hysteresis sequence. a, The I-V hysteresis follows the order of HRS→LRS→LRS→HRS. b, The I-V hysteresis follows the order of HRS→LRS→HRS→LRS.

The following description has been added in the revised manuscript (on Page 14, Line 295).

Compared with previously reported bipolar memristors (Supplementary Section 12), the key parameters (such as storage medium thickness, rectification ratio, ON/OFF ratio, switching voltage, ON-state current density, and so on) of our molecular memristors reach a comparable level except for the endurance stability.

The following description has been added in the revised Supplementary Information (in Section 12, on Page S35).

Section 12. Comparison of the memristive performance with previously reported memristors

Here, we focus on bipolar memristors and divide them into two categories according to the sequence of the I-V hysteresis loops (as shown in Figure R10, also in Supplementary Fig. 12): following the order of (1) HRS→LRS→LRS→HRS (HLLH) or (2) HRS→LRS→HRS→LRS (HLHL), where HRS and LRS are the abbreviations of high resistance state (i.e., OFF state) and low resistance state (i.e., ON state), respectively.

Most HLHL switching phenomena are found in perovskite-based memristors, in which switching is ascribed to the barrier modulation induced by the polarization of the mobile ions in the perovskites or the polarization of the ferroelectric domains. This kind of memristors also exhibits the “switchable diode effect”. Therefore, they can work as both forward and reverse diodes, which depend on the polarization direction. Such HLHL behaviour and switchable diode effect are rarely reported in molecular/organic memristors but they occur in our devices. Compared with previously reported

perovskite-based memristors the performance of our devices is comparable in terms of the rectification ratio, ON/OFF ratio, switching voltage, and retention stability. On the other hand, the ON-state current density of our devices is higher than that of most perovskite-based memristors. Moreover, there is no need to conduct a poling process to activate the memristive switching behaviour in the molecular devices in our work. Poling refers to applying a higher voltage over a period of time to create polarization in perovskite materials. For the molecular devices in our work, it is supposed that the ultrahigh electric field (for example, the average electric field is about 1.67 MV/cm at 1 V) immediately polarizes the mobile ions through the ultrathin molecular hybrid layer. However, compared to the HLLH memristors listed in Supplementary Table 3, the endurance capability of our devices still needs to be enhanced.

Supplementary Table 3 | Comparison of different techniques for creating sub-10 nm thick molecular devices

Device structure ^a	Thickness (nm)	Hysteresis order ^b	Forward RR@±V ^c	Reverse RR@±V	OFF/ON ratio@V	Switching voltage (V)	ON-state current density (A/cm ²)	Forming required? ^d	Retention (s)	Endurance (cycles)	Ref.
Ag/SAM/GaO _x /EGaIn	~2	HLLH	-	2.5×10 ⁴ @1.0 V ^e	6.7×10 ³ @-0.3 V	-0.89	~10 ⁻² @-0.5 V	No	1.2×10 ⁴	10 ⁶	79
ITO/Au NPs/ [Ru(L) ₃](PF ₆) ₂ /ITO	10-100	HLLH	-	-	~10 ⁵ @0.3 V	~0.52	~10 ³ @0.3	No	>10 ⁶	~10 ¹²	80
Pt/PA-TsOH/Pt	90	HLLH	-	-	>10 ⁵ @0.1 V	~7.0	-	No	10 ⁵	>10 ³	81
ITO/Poly-1 ⁴⁺ /Al	100	HLLH	-	-	~2.0×10 ² @1.0 V	3.4	~10 ⁻² @1.0	No	1.2×10 ⁴	500	82
Ag/MoS ₂ /MoO _x /Ag	50-100	HLLH	-	-	~10 ⁶ @~0 V	0.1-0.2	~10 @0.01	No	~10 ⁴	3.0×10 ⁴	83
ITO/CsPbBr ₃ ODs/PMMA/Ag	~800	HLLH	-	-	~10 ³ @0.5 V	~1.0	-	Electro- forming	10 ⁵	10 ⁴	84
Pt/BCZT/HAO/Au	50	HLHL	~10 ⁴ @0.1 V	~10 ⁴ @0.1 V	~10 ⁴ @-0.1 V	~0.2	~10 ⁻⁵ @-0.1 V	Not specified	10 ⁴	60	85
FTO/BiFeO ₃ /C-AFM	250	HLHL	~10 ⁴ @2.0 V	~7 @4.0 V	~7.0×10 ³ @2.0 V	~1.0	~10 ⁻² @2.0 V	Poling @±9V	10 ³	-	86
STO/BiFeO ₃ /C-AFM	240	HLHL	-	~100 @1.0 V	~10 ³ @-2.0 V	~-1.0	~10 ⁻² @-2.0 V	Poling @±10V	10 ⁶	-	67
STO/SRO/BiFeO ₃ / C-AFM	40	HLHL	~500 @0.6 V	~200 @0.6 V	~753 @0.5 V	~2.0	-	Poling @±3.5V	10 ³	-	59

ITO/PEDOT:PSS/ MAPbI ₃ /Au	300	HLHL	~200 @1.0 V	~300 @1.0 V	>10 ³ @-1.0 V	~-1.0	~10 ⁻² @-1.0 V	Poling @±2.5V	-	-	61
SRO/BiFeO ₃ (Ca)/ C-AFM	100	HLHL	-	~10 ³ @10 V	~10 ³ @-2.0 V	~-5.0	-	Poling @±12V	-	-	62
PPA60/CuPc/C ₆₀ /Au	6	HLHL	~400 @0.5 V	~900 @0.5 V	~3×10 ³ @0.5 V	~-0.8	>10 @0.5 V	No	2000	100	This work

a, The Red part indicates the storage medium.

b, According to the sequence of the I-V hysteresis loops, bipolar memristors are divided into two categories: following the order of (1) HRS→LRS→LRS→HRS (HLLH) or (2) HRS→LRS→HRS→LRS (HLHL), where HRS and LRS are the abbreviations of high resistance state (i.e., OFF state) and low resistance state (i.e., ON state), respectively.

c, RR= Rectification Ratio.

d, The “forming” in filamentary memristors refers to “electroforming”, while in ferroelectric-type memristors it means “poling”.

e, In this work, the rectifying behaviour arises from the extremely asymmetrical I-V hysteresis loops rather than the “switchable diode effect” that usually occurs in ferroelectric-type memristors.

Related references in the Supplementary Information:

59. Hong, S., Choi, T., Jeon, J.H., Kim, Y., Lee, H., Joo, H.Y., Hwang, I., Kim, J.S., Kang, S.O., Kalinin, S.V. Large resistive switching in ferroelectric BiFeO₃ nano-island based switchable diodes. *Adv. Mater.* **25**, 2339-2343 (2013).
61. Xiao, Z., Yuan, Y., Shao, Y., Wang, Q., Dong, Q., Bi, C., Sharma, P., Gruverman, A., Huang, J. Giant switchable photovoltaic effect in organometal trihalide perovskite devices. *Nat. Mater.* **14**, 193-198 (2015).
62. Yang, C.H., Seidel, J., Kim, S.Y., Rossen, P.B., Yu, P., Gajek, M., Chu, Y.H., Martin, L.W., Holcomb, M.B., He, Q., Maksymovych, P., Balke, N., Kalinin, S.V., Baddorf, A.P., Basu, S.R., Scullin, M.L., Ramesh, R. Electric modulation of conduction in multiferroic Ca-doped BiFeO₃ films. *Nat. Mater.* **8**, 485-493 (2009).
67. Jiang, A.Q., Wang, C., Jin, K.J., Liu, X.B., Scott, J.F., Hwang, C.S., Tang, T.A., Lu, H.B., Yang, G.Z. A resistive memory in semiconducting BiFeO₃ thin-film capacitors. *Adv. Mater.* **23**, 1277-1281 (2011).
79. Han, Y., Nickle, C., Zhang, Z., Astier, H., Duffin, T.J., Qi, D., Wang, Z., Del Barco, E., Thompson, D., Nijhuis, C.A. Electric-field-driven dual-

functional molecular switches in tunnel junctions. *Nat. Mater.* **19**, 843-848 (2020).

80. Goswami, S., Matula, A.J., Rath, S.P., Hedstrom, S., Saha, S., Annamalai, M., Sengupta, D., Patra, A., Ghosh, S., Jani, H., Sarkar, S., Motapothula, M.R., Nijhuis, C.A., Martin, J., Goswami, S., Batista, V.S., Venkatesan, T. Robust resistive memory devices using solution-processable metal-coordinated azo aromatics. *Nat. Mater.* **16**, 1216-1224 (2017).
81. Hu, B., Zhu, X., Chen, X., Pan, L., Peng, S., Wu, Y., Shang, J., Liu, G., Yan, Q., Li, R.W. A multilevel memory based on proton-doped polyazomethine with an excellent uniformity in resistive switching. *J. Am. Chem. Soc.* **134**, 17408-17411 (2012).
82. Cui, B.-B., Mao, Z., Chen, Y., Zhong, Y.-W., Yu, G., Zhan, C., Yao, J. Tuning of resistive memory switching in electropolymerized metallopolymeric films. *Chem. Sci.* **6**, 1308-1315 (2015).
83. Bessonov, A.A., Kirikova, M.N., Petukhov, D.I., Allen, M., Ryhanen, T., Bailey, M.J. Layered memristive and memcapacitive switches for printable electronics. *Nat. Mater.* **14**, 199-204 (2015).
84. Yen, M.C., Lee, C.J., Liu, K.H., Peng, Y., Leng, J., Chang, T.H., Chang, C.C., Tamada, K., Lee, Y.J. All-inorganic perovskite quantum dot light-emitting memories. *Nat. Commun.* **12**, 4460 (2021).
85. Silva, J., Silva, J.P.B., Sekhar, K., Pereira, M., Gomes, M. Impact of the ferroelectric layer thickness on the resistive switching characteristics of ferroelectric/dielectric structures. *Appl. Phys. Lett.* **113**, 102904 (2018).
86. Zhang, L., Chen, J., Cao, J., He, D., Xing, X. Large resistive switching and switchable photovoltaic response in ferroelectric doped BiFeO₃-based thin films by chemical solution deposition. *J. Mater. Chem. C* **3**, 4706-4712 (2015).

In all, I find their devices and some of the concepts very interesting (i.e., role of mobile ions) but the authors should refrain from overhyping, provide proper context and acknowledge relevant prior work, and give a balanced discussion regarding the benefit but also disadvantages of their fabrication method before I can support publication.

Our response:

We appreciate these valuable comments and your suggestions on giving a balanced discussion of relevant prior work and different manufacturing techniques, which has helped us to improve the quality of our work. Please refer to our **Response (3.1)** for details.

REVIEWERS' COMMENTS

Reviewer #1 (Remarks to the Author):

Zhu et al have comprehensively addressed the referee concerns; the new material in the main text and SI have significantly improved the quality of the work and the referee-only material is very convincing. Recommend publish as is.

Reviewer #3 (Remarks to the Author):

The authors went through extraordinary efforts to improve their manuscripts. They have addressed all my questions and followed all suggestions which improved their manuscript considerably. Especially the balanced introduction, comparisons to other techniques, and additional information on statistics (and many more) ensure that their work will serve as an important benchmark for future works. I fully support publication now.

Editorial Note: Parts of this peer review file have been redacted as indicated to remove third-party material where no permission to publish could be obtained.

REVIEWER COMMENTS

Reviewer #1 (Remarks to the Author):

Zhu and coworkers present a very interesting new field-controlled memristive switching molecular device with integrated microfluidics using conductive polymers, that moves ion-controlled switching closer to wafer scale. It also functions as a photomultiplier. It is clever and novel science, and suitable for publication in Nat Comm following some corrections:

1. Can the authors be confident that silver "leaching" via migration/diffusion of Ag nanoparticles does not contribute to switching in a filamentation type mechanism?
2. The notion of coupled ionic-electronic effects is not very helpful in understanding the mechanism. It would be great if future DFT studies could clarify the mechanism but until then I would urge caution in how the proposed mechanism is presented.
3. On a related point, and my largest concern here, is lack of molecular scale characterisation of the polymer/molecule interface - Raman, microscopy and conductivity measurements look to be very careful and show high quality film but it would improve the usefulness of the m/s to present more atomic scale characterisation, assisted by modelling as necessary, to demonstrate how the interface promotes the ion diffusion mechanism. If not, then control materials should be proposed, both positive and negative, to benchmark the findings for PDOT:PSS as the source of the "right" mobile ions for the switching.

Minor thing:

The paper should be carefully proofread to avoid errors and improve readability - e.g., "inhere a large range of" in the first sentence of the Abstract, the unnecessary "over the years" in the second sentence.

Reviewer #2 (Remarks to the Author):

In this manuscript, the authors report on a integrated molecular diode-like devices that exhibit

reconfigurable behavior. More specifically, an ionic reservoir can modify the behavior of the device from a diode to a photodiode. The device behavior is explained in detail. Nevertheless, the manuscript does not present any computational/processing aspect close or beyond the state-of-art. Therefore, I believe that the manuscript is suitable for a more specialized journal.

Reviewer #3 (Remarks to the Author):

The manuscript by Li et al. with the title “Process-Programmable Molecular Junctions Switch between Photomultiplication and Memristive Behaviour in On-Chip Integrated Devices” demonstrate memristive behavior of a new type of molecular junction. They report a new type of molecular electronic device which is very interesting and deserves publication in Nat Commun after the authors have addressed the following points.

- “due to the lack of damage-free micro-nano fabrication techniques, it is still a huge challenge to integrate and control the ionic and electronic functions within the molecular devices in parallel at wafer-scale, which is crucial to develop novel and practical functional molecular devices.” This statement is correct for single-molecule junctions, but not for large area junctions. There are quite a few established fabrication techniques that have withstood the test of time , some are even commercially available (DOI: 10.1088/0953-8984/28/9/094011 ; <https://dx.doi.org/10.1021/jacs.9b12424>; but also the groups of Cahen and Lacroix make highly stable molecular junctions on large scales, group of Chiechi reported impressive ways to fabricate functional molecular devices (including memory) with very high stabilities (see doi: 10.1063/5.0050667 for an overview and citations therein). The introduction should be toned down and include a more balanced discussion

- The authors state ”it is reasonable to claim that the ensembles of CuPc/C60 molecules fall into the molecular scale”, but in their explanation of the mechanism of they fall back on semiconductor physics which goes against this statement. Schottky junctions, p-n junctions, band bending, can only form when depletion layers are present (which usually far exceed molecular dimensions), such depletion layers cannot form in molecules. In molecular junctions, HOMO/LUMO levels can shift (energy level alignment) for instance, and since there is only one HOMO/LUMO per molecules, band bending is not possible. The proposed model and explanations suggest that their devices are not truly molecular in nature as ions from the electrode material play a crucial role. The authors should modify their manuscript accordingly

After reading the bold introduction, my expectations were high but ...:

- Lack of statistics. No yields, error bars (reproducibilities) etc are reported. Normally heatmaps, tables with statistics (number of samples, number of devices, number of measurements etc), (log)average curves with (log)standard deviations etc should be all given but not reported here.

- Stability. No endurance, retention, or demonstration of non-volatility are given.

- Fig 2b shows that their technique may be very difficult to scale. I find this not a major issue, but this should be mentioned and compared to other fabrication techniques

- How well do their devices perform with respect to other devices? For instance, in their ref 7 the on/off ratios are higher, but there are more examples of molecular memristors in the literature

In all, I find their devices and some of the concepts very interesting (i.e., role of mobile ions) but the authors should refrain from overhyping, provide proper context and acknowledge relevant prior work, and give a balanced discussion regarding the benefit but also disadvantages of their fabrication method before I can support publication.

Response to the comments of Reviewer #1

Reviewer #1 (Remarks to the Author):

Comments: Zhu and coworkers present a very interesting new field-controlled memristive switching molecular device with integrated microfluidics using conductive polymers, that moves ion-controlled switching closer to wafer scale. It is also functions as photomultiplier. It is clever and novel science, and suitable for publication in Nat Comm following some corrections:

Our response:

Thank you very much for your efforts on our manuscript. We greatly appreciate your advice and comments which are very helpful for improving the quality of this work. We have revised the manuscript according to your suggestions. The revised parts are marked in red. Our response is presented point-to-point in the following.

Question (1.1) *Can the authors be confident that silver "leaching" via migration/diffusion of Ag nanoparticles does not contribute to switching in a filamentation type mechanism?*

Response (1.1)

We appreciate this comment a lot. Since there is a strong relationship among the Reviewer's first three questions, i.e., all of them focus on the memristive mechanism, we would like to provide a comprehensive reply in our Response (1.1) to clarify the working mechanism, and then make additions to our Response (1.2) and Response (1.3). The phenomena observed in our work, i.e., the I-V hysteresis order (see Figure R1a and d), the linear fitting of double logarithmic I-V plot (see Figure R1b, c, e and f), the switchable diode effect (see Figure R2), and the I-V hysteresis shrink during light illumination (see Figure R3), are usually found in perovskite-based barrier-type memristors, which indicates that hysteretic behaviour of our molecular devices originates from ion polarization-modulated charge injection, rather than the formation/rupture of filaments. On the other hand, we conduct a series of control experiments (i.e., replacing the polymer bottom electrode (see Figure R4a-c), replacing the molecular storage medium (see Figure R4d-f), and washing the etched PEDOT:PSS:Ag NWs electrode over a long time (see Figure R5)), and all the results indicate that (1) the etched PEDOT:PSS:Ag NWs electrode is the only source of ions for resistive switching and (2) the hybrid molecular layer is not the decisive factor for the hysteretic behaviour in our work.

A detailed discussion about the memristive mechanism of our molecular memristor is presented below, which has also been added to the revised Supplementary Information (see Section 10, on Page S26).

Section 10. The memristive mechanism

S10.1 Filamentary vs. barrier modulation

To elucidate the working mechanism, we first compare the I-V hysteresis features of our memristors to those of conventional filamentary memristors, as shown in Figure R1 (also in Supplementary Fig. 12). We fabricated a stacked structure of Ni/SiO₂(1 nm)/Al₂O₃(1 nm)/NiFe, which exhibits a typical filamentation-type hysteresis behaviour^{57,58} (Figure R1a, also in Supplementary Fig. 12a) owing to the formation and rupture of conductive filaments. In this situation the sequence of the I-V hysteresis follows HRS→LRS→LRS→HRS, where HRS and LRS are the abbreviations of the high resistance state (i.e., OFF state) and low resistance state (i.e., ON state), respectively. Therefore, the memristor can maintain the ON/OFF state when the bias polarity is switched. However, for the molecular devices in our work (as shown in Figure R1, also in Supplementary Fig. 12d), the I-V hysteresis follows the order HRS→LRS→HRS→LRS; i.e., the resistance state changes whenever the bias polarity switches, which is consistent with the behaviour of memristors that rely on a Schottky barrier modulation in perovskite-based devices, such as BiFeO₃ (BFO)⁵⁹, BaTiO (BTO)⁶⁰, CH₃NH₃PbI₃ (MAPbI₃)⁶¹, and so on. For this kind of perovskite memristors, the observed hysteresis is generally accepted to be caused by ion migration-induced barrier modulation, which is supported by experimental results^{60,61,62} and numerical simulations⁶³. Under an electric field, mobile ions drift in the perovskite layer and accumulate near the electrode surfaces, leading to p-doping in the perovskite near the anode and n-doping near the cathode. Similarly, a reverse bias can flip the doping configuration by forcing positive and negative ions to drift in the opposite direction. The self-doping effect can influence the electrode/perovskite barrier width and therefore increase or decrease the carrier injection capability, resulting in memristive hysteresis. In perovskite materials, the mobile ions originate from lattice defects at the grain boundaries, for example, positively charged I vacancies and negatively charged Pb and MA vacancies in MAPbI₃⁶¹.

Additional evidence for clarifying the mechanism is provided by fitting the double logarithmic I-V plot with a power law⁶⁴, as shown in Figure R1 (also in Supplementary Fig. 12). For filamentation-type memristors, the ON state is attributed to the formation of conductive filaments between the two electrodes, which causes a “short circuit”. Therefore, the charge transport of the On-state is governed by Ohmic conduction, and the corresponding I-V plots of the LRS for both negative and positive voltages are characterized by Ohmic behaviour ($I \sim V$)^{65,66}, i.e., the slope is close to 1 (as shown in Figure R1b and 1c, also in Supplementary Fig. 12b and 12c). However, for the molecular devices in our work (as shown in Figure R1e and 1f, also in Supplementary Fig. 12e and 12f) the slopes of the

On-state I-V plots at negative and positive voltages are calculated to be 2.1 and 3.2, respectively, which deviate significantly from Ohmic behaviour. Instead, the linear fitting matches well with perovskite-based memristive switching⁶⁰.

Figure R1 | Two types of bipolar memristors according to the I-V hysteresis sequence. **a**, I-V hysteresis follows the order of HRS→LRS→LRS→HRS. The example shown here is based on a stacked structure of Ni/SiO₂(1 nm)/Al₂O₃(1 nm)/NiFe. **b** and **c**, Double logarithmic fittings of the negative part (**b**) and positive part (**c**) of the I-V plot shown in **a**. **d**, I-V hysteresis follows the order of HRS→LRS→HRS→LRS. The example shown here is based on our PPA60/CuPc(3 nm)/C₆₀(3 nm)/Au. **e** and **f**, Double logarithmic fittings of the negative part (**e**) and positive part (**f**) of the I-V plot shown in **d**.

S10.2 Switchable diode effect

The switchable diode effect that usually appears in interface-type perovskite memristors^{59, 67} is also observed in our molecular devices (as shown in Figure R2, also in Supplementary Fig. 13), which has not yet been reported in filamentation-type memristors as far as we know. For the virgin state, the current is very small and nearly symmetric over a small bias range (Figure R2a, also in Supplementary Fig. 13a); however, when the sweep starts from a higher negative voltage (-1.2 V here), the current increases rapidly in the negative voltage region but increase slowly in the positive voltage region, indicating a reverse diode behaviour (Figure R2b, also in Supplementary Fig. 13b); whereas, when the sweep starts from a higher positive voltage (+1.2 V here), the current shows a forward diode (Figure

R2c, also in Supplementary Fig. 13c). The mechanism of the switchable diode effect in perovskite devices is still under debate: some researchers claim that it originates from the polarization of mobile ions^{61, 62, 68}; while some others claim that it is governed by the polarization of ferroelectric domains^{59, 69}. However, both of them agree that the polarized charges (ions or charged centres) under the applied electric field can modify the electrode/perovskite Schottky barriers, leading to switchable diode behaviour. It should be pointed out that in our case both CuPc and C₆₀ are not ferroelectric materials.

Figure R2 | Voltage-polarity-dependent rectifying behaviour. **a**, Virgin state extracted from Figure 4a1. **b** and **c**, Reverse diode (**b**) and forward diode (**c**) behaviours extracted from Figure 4a2. **d**, Single sweep from PPA60 bottom electrode to Au tubular top electrode, resulting in a diode-like rectifying behaviour with negative voltage as the forward bias. **e**, Single sweep from Au tubular top electrode to PPA30 bottom electrode, resulting in a diode-like rectifying behaviour with positive voltage as the forward bias. The bottom PPA60 electrode is grounded.

S10.3 Photoresponse of the molecular memristors with I-V hysteresis

As shown in Figure R3 (also in Supplementary Fig. 14), compared to the dark condition, the I-V hysteresis almost disappears under illumination. The shrinkage of the I-V hysteresis under illumination

has also been reported in perovskite-based solar cells and photodetectors due to ion-involved phenomena^{69, 70}. Most of the mobile ions drift toward the contacts under a high external electric field, and in turn, the ion accumulation impacts the electron and hole distributions (i.e., ion accumulation-induced doping effect), which is deleterious to the performance of the devices in the photovoltaic and light detection fields. Owing to the presence of a very large electron/hole concentration at the interface (where photogenerated holes/electrons would be extracted), these accumulated ions at the electrode interfaces can act as electron-hole recombination centers (i.e., traps) to enhance the interfacial charge recombination during the collection of photogenerated carriers⁷¹. As a consequence, the barrier modulation effect induced by ion accumulation is weakened, hence, the photocurrent is reduced⁷² and the I-V hysteresis reduces under illumination after polarization. Furthermore, the presence of ions at the interfaces can lead to chemical reactions at the contacts, which could also modify the extraction properties. Back to our case, it is supposed that both the enhanced surface recombination and chemical reaction contribute to the shrinkage of the I-V hysteresis. Because after turning off the light, the I-V hysteresis in the dark is severely reduced, especially for the negative part, which is possibly ascribed to the lack of Ag ions owing to the photoreduction of Ag ions with the help of photogenerated electrons.

Figure R3 | Photoresponse of the molecular device based on PPA60/CuPc (5 nm)/C₆₀ (5 nm)/Au tube. a, I-V characteristics in dark. b, I-V characteristics under illumination. c, I-V characteristics in the dark again.

S10.4 Determining the source of the mobile ions

All the abovementioned phenomena indicate that our memristive switching performance stems from the ion migration-induced barrier modulation rather than the formation/rupture of filaments. Usually, interface-type memristors have been widely reported for perovskite-based materials, in which the mobile ions originate from lattice defects. However, in our work, a series of experiments point out that the mobile ions are provided by the etched PEDOT:PSS:Ag NWs, i.e., they are not from the organic hybrid layer CuPc/C₆₀. As shown in Figure R4a and 4b (also in Supplementary Fig. 15a and 15b),

when the etched PEDOT:PSS:Ag NWs electrode is replaced by an ITO or Ag film, the I-V hysteresis is negligible. However, when the Ag film electrode is etched for a short time (~2 s) with the same etching solution, the hysteretic behaviour occurs again, but it is very unstable (Figure R4c, also in Supplementary Fig. 15c). This result directly implies that the etched Ag film can provide ions but cannot control the ions as effectively as etched PEDOT:PSS:Ag NWs during the repeated sweeping. Moreover, when CuPc is replaced by H₂PC, a similar I-V hysteresis is also observed (Figure R4d, also in Supplementary Fig. 15d), which not only indicates that the coordinated metallic ions Cu(II) in CuPc do not contribute to the hysteretic phenomenon but also demonstrates that the etched PEDOT:PSS:Ag NWs electrode indeed matters. We also prepared PPA60-based devices with a single CuPc or C₆₀ layer as the active material (Figure R4e and 4f, also in Supplementary Fig. 15e and 15f), and find that both PPA60/CuPc(3 nm)/Au and PPA60/C₆₀(3 nm)/Au exhibit I-V hysteresis. However, the hysteresis loops are much weaker than that of PPA60/CuPc(3 nm)/C₆₀(3 nm)/Au. This observation emphasizes that the ultrathin CuPc/C₆₀ hybrid layer is very important to achieve excellent resistance switching performance, and also implies that the inserted molecules are not the core factor causing the hysteresis, but instead the etched PEDOT:PSS:Ag NWs electrode.

Figure R4 | Determining the origin of the mobile ions. a, I-V characteristics of the molecular device based on ITO/CuPc (3 nm)/C₆₀ (3 nm)/Au tube. **b** and **c**, I-V characteristics of the molecular devices

based on Ag (40 nm)/CuPc (3 nm)/C₆₀ (3 nm)/Au tube without etching the Ag film (b) and with etching the Ag film for a short time (c). d-f, I-V characteristics of the molecular devices based on PPA60/H₂Pc (3 nm)/C₆₀ (3 nm)/Au tube (d), PPA60/CuPc (3 nm)/Au tube (e), and PPA60/C₆₀ (3 nm)/Au tube (f), respectively.

S10.5 Transition between photomultiplication photodiodes and bipolar memristors

Another experimental evidence is that when increasing the water washing time (from 10 s to 10 min) of PPA60 after wet etching (i.e., before molecule deposition), the corresponding molecular devices act as photomultiplication photodiodes again, and the hysteresis behaviour is suppressed (Figure R5, also in Supplementary Fig. 16). This is because the longer washing procedure partly removes the mobile ions, leading to a decrease in ion concentration and a reduction of the PPA60/CuPc barrier height, both of which make ion transport difficult. This phenomenon further supports the fact that the etched PEDOT:PSS:AgNWs film is the source of mobile ions. In fact, PEDOT:PSS has been extensively used as an “ion reservoir” to store ions. We use the Raman spectrum and light absorption coefficient profile to characterize the PEDOT:PSS:AgNWs film before and after etching, and find that after etching there are no metallic Ag related signals (see Figure 2f and g in the manuscript). However, the XPS spectrum exhibits a clear Ag signal (see Figure 2h in the manuscript), and the relative atomic content increases with the etching time. If the Ag signal comes from metallic Ag, then the content should decrease as the etching time is prolonged. Obviously, this did not happen. Based on these results, we conclude that Ag nanowires are completely gone after etching and the Ag ions are adsorbed on the etched PEDOT:PSS. Moreover, N is also detected, and the relative atomic content increases with the etching time, which stems from the etching component HNO₃. The mixed etching solution was H₃PO₄ (3):HNO₃ (3):CH₃COOH (23): H₂O (1). In fact, both Ag₃PO₄ and CH₃COOAg exist in solid forms, which may be the reason why element P is not detected.

Figure R5 | Transition between photomultiplication photodiodes and bipolar memristors. a, I-V characteristics of the molecular device based on PEDOT:PSS:AgNWs film wet-etched for 30 s and washed with water for 10 s. **b,** I-V characteristics of the molecular device based on PEDOT:PSS:AgNWs film wet-etched for 60 s and washed with water for 10 s. **c,** I-V characteristics of molecular device based on PEDOT:PSS:AgNWs film wet-etched for 60 s and washed with water for 10 min.

In summary, we have clarified that our memristive switching performance is attributed to ion migration-induced barrier modulation (instead of the formation/rupture of filaments), and we have provided evidence that the mobile ions originate from the etched PEDOT:PSS film (rather than the organic hybrid layer CuPc/C₆₀).

Related references in the Supplementary Information:

57. Lee, B.-H., Bae, H., Seong, H., Lee, D.-I., Park, H., Choi, Y.J., Im, S.-G., Kim, S.O., Choi, Y.-K. Direct observation of a carbon filament in water-resistant organic memory. *ACS Nano* **9**, 7306-7313 (2015).
58. Raeis Hosseini, N., Lee, J.-S. Resistive switching memory based on bioinspired natural solid polymer electrolytes. *ACS Nano* **9**, 419-426 (2015).
59. Hong, S., Choi, T., Jeon, J.H., Kim, Y., Lee, H., Joo, H.Y., Hwang, I., Kim, J.S., Kang, S.O., Kalinin, S.V. Large resistive switching in ferroelectric BiFeO₃ nano-island based switchable diodes. *Adv. Mater.* **25**, 2339-2343 (2013).
60. Chen, A., Zhang, W., Dedon, L.R., Chen, D., Khatkhatay, F., MacManus-Driscoll, J.L., Wang, H., Yarotski, D., Chen, J., Gao, X., Martin, L.W., Roelofs, A., Jia, Q. Couplings of Polarization with Interfacial Deep Trap and Schottky Interface Controlled Ferroelectric Memristive Switching. *Adv. Funct. Mater.* **30**, 2000664 (2020).
61. Xiao, Z., Yuan, Y., Shao, Y., Wang, Q., Dong, Q., Bi, C., Sharma, P., Gruverman, A., Huang, J. Giant switchable photovoltaic effect in organometal trihalide perovskite devices. *Nat. Mater.* **14**, 193-198 (2015).
62. Yang, C.H., Seidel, J., Kim, S.Y., Rossen, P.B., Yu, P., Gajek, M., Chu, Y.H., Martin, L.W., Holcomb, M.B., He, Q., Maksymovych, P., Balke, N., Kalinin, S.V., Baddorf, A.P., Basu, S.R., Scullin, M.L., Ramesh, R. Electric modulation of conduction in multiferroic Ca-doped BiFeO₃ films. *Nat. Mater.* **8**, 485-493 (2009).
63. Sajedi Alvar, M., Blom, P.W.M., Wetzelaer, G.A.H. Space-charge-limited electron and hole currents in hybrid organic-inorganic perovskites. *Nat. Commun.* **11**, 4023 (2020).
64. Lim, E., Ismail, R. Conduction Mechanism of Valence Change Resistive Switching Memory: A Survey. *Electronics* **4**, 586-613 (2015).
65. Liu, L., Li, Y., Huang, X., Chen, J., Yang, Z., Xue, K.H., Xu, M., Chen, H., Zhou, P., Miao, X. Low-Power Memristive Logic Device Enabled by Controllable Oxidation of 2D HfSe₂ for In-Memory Computing. *Adv. Sci.* **8**, 2005038 (2021).
66. Li, Y., Zhou, Y.-X., Xu, L., Lu, K., Wang, Z.-R., Duan, N., Jiang, L., Cheng, L., Chang, T.-C., Chang, K.-C. Realization of functional complete stateful Boolean logic in memristive crossbar. *ACS Appl. Mater. Interfaces* **8**, 34559-34567 (2016).
67. Jiang, A.Q., Wang, C., Jin, K.J., Liu, X.B., Scott, J.F., Hwang, C.S., Tang, T.A., Lu, H.B., Yang, G.Z. A

- resistive memory in semiconducting BiFeO₃ thin-film capacitors. *Adv. Mater.* **23**, 1277-1281 (2011).
68. Lee, J.H., Jeon, J.H., Yoon, C., Lee, S., Kim, Y.S., Oh, T.J., Kim, Y.H., Park, J., Song, T.K., Park, B.H. Intrinsic defect-mediated conduction and resistive switching in multiferroic BiFeO₃ thin films epitaxially grown on SrRuO₃ bottom electrodes. *Appl. Phys. Lett.* **108**, 112902 (2016).
69. Choi, T., Lee, S., Choi, Y.J., Kiryukhin, V., Cheong, S.W. Switchable ferroelectric diode and photovoltaic effect in BiFeO₃. *Science* **324**, 63-66 (2009).
70. Kwon, K.C., Hong, K., Van Le, Q., Lee, S.Y., Choi, J., Kim, K.B., Kim, S.Y., Jang, H.W. Inhibition of ion migration for reliable operation of organolead halide perovskite-based Metal/Semiconductor/Metal broadband photodetectors. *Adv. Funct. Mater.* **26**, 4213-4222 (2016).
71. Lan, D. The physics of ion migration in perovskite solar cells: Insights into hysteresis, device performance, and characterization. *Prog. Photovolt. Res. Appl.* **28**, 533-537 (2020).
72. Lan, C., Zou, H., Wang, L., Zhang, M., Pan, S., Ma, Y., Qiu, Y., Wang, Z.L., Lin, Z. Revealing Electrical-Poling-Induced Polarization Potential in Hybrid Perovskite Photodetectors. *Adv. Mater.* **32**, 2005481 (2020).

Accordingly, the following description has been added to the revised manuscript (on Page 12, Line 237).

The resistance switching in the PPA60/CuPc/C₆₀/Au tube structure is further explored. As described in detail in Supplementary Section 10, the I-V hysteresis order and linear fitting (Figure R1, also in Supplementary Fig. 12), the switchable diode effect (Figure R2, also in Supplementary Fig. 13), and the hysteresis reduction during light illumination (Figure R3, also in Supplementary Fig. 14) suggest that the hysteretic behaviour of our molecular devices is analogous to that of barrier modulation-related perovskite memristors, instead of conventional filamentary memristors. For memristors based on perovskite materials, e.g., BiFeO₃ films with defects such as bismuth and oxygen deficiencies (acting as mobile ions)^{34, 35}, the hysteresis is attributed to the Schottky barrier modulation induced by ion polarization in the perovskite storage medium.

Related references in the manuscript:

34. Yang, C.H., Seidel, J., Kim, S.Y., Rossen, P.B., Yu, P., Gajek, M., Chu, Y.H., Martin, L.W., Holcomb, M.B., He, Q., Maksymovych, P., Balke, N., Kalinin, S.V., Baddorf, A.P., Basu, S.R., Scullin, M.L., Ramesh, R. Electric modulation of conduction in multiferroic Ca-doped BiFeO₃ films. *Nat. Mater.* **8**, 485-493 (2009).
35. Xiao, Z., Yuan, Y., Shao, Y., Wang, Q., Dong, Q., Bi, C., Sharma, P., Gruverman, A., Huang, J. Giant switchable photovoltaic effect in organometal trihalide perovskite devices. *Nat. Mater.* **14**, 193-198 (2015).

Question (1.2) *The notion of coupled ionic-electronic effects is not very helpful in understanding the mechanism. It would be great if future DFT studies could clarify the mechanism but until then I would urge caution in how the proposed mechanism is presented.*

Response (1.2)

We greatly appreciate your advice and concern. As mentioned in our Response (1.1) to your first **Question**, the mechanism of the hysteretic behaviour in our molecular devices is proposed to be the ion accumulation-enhanced charge injection. One interesting point is, in fact, that the HRS→LRS→HRS→LRS hysteresis sequence, the switchable diode effect, and the hysteresis reduction upon light illumination are usually found in perovskite-based memristors/solar cells, which are attributed to the Schottky barrier modulation induced by the ion polarization in the perovskites. All of these phenomena occur in our molecular devices; therefore, we rely on existing phenomena in the field of perovskite memristors to explain the working mechanisms in our devices.

***Question (1.3)** On a related point, and my largest concern here, is lack of molecular scale characterisation of the polymer/molecule interface - Raman, microscopy and conductivity measurements look to be very careful and show high quality film but it would improve the usefulness of the m/s to present more atomic scale characterisation, assisted by modelling as necessary, to demonstrate how the interface promotes the ion diffusion mechanism. If not, then control materials should be proposed, both positive and negative, to benchmark the findings for PEDOT:PSS as the source of the "right" mobile ions for the switching.*

Response (1.3)

We have taken your advice and carried out a series of control experiments to clarify the working mechanisms. As illustrated in the Response (1.1) to your first **Question**, we replaced the etched PEDOT:PSS:Ag NWs bottom electrode with inorganic material, replaced the CuPC/C₆₀ hybrid layer with other components, and washed the etched PEDOT:PSS:Ag NWs for extended periods. According to the experimental results, we conclude that (1) the CuPC/C₆₀ hybrid layer is not the origin of the hysteretic behaviour, and (2) the etched PEDOT:PSS:Ag NWs electrode leads to the hysteresis phenomenon.

Alvar *et al.*, [*Nat. Commun.* 11, 4023 (2020)] used the electronic-ionic drift-diffusion model (via MATLAB) to simulate the hysteresis in the archetypical perovskite methyl ammonium lead iodide (MAPbI₃), and revealed that the coupling between ion migration and electron/hole injection gave rise to the hysteresis in the order of HRS→LRS→HRS→LRS (as shown in Figure R6). Although the system described by their model is much simpler than ours, it provides a valuable reference and supports the interpretation in our work.

[Redacted]

Figure R6 | Current density-voltage characteristics of a hole-only device at 275 K. The current density-voltage characteristics (solid lines) are simulated with the same set of parameters under two different conditions for ionic charges: mobile positive ions and a uniform density of immobile negative ions **(a)** and mobile negative ions and immobile positive ions **(b)**. Source: *Sajedi Alvar M, Blom PWM, Wetzelaer GAH. Space-charge-limited electron and hole currents in hybrid organic-inorganic perovskites. Nat Commun 11, 4023 (2020).*

Question (1.4) Minor thing:

The paper should be carefully proofread to avoid errors and improve readability - e.g., "inhere a large range of" in the first sentence of the Abstract, the unnecessary "over the years" in the second sentence.

Response (1.4)

Thank you very much for your suggestion. In the revised manuscript, we carefully polished our language and hope the manuscript is more readable now.

Response to the comments of Reviewer #2

Reviewer #2 (Remarks to the Author):

Comments: In this manuscript, the authors report on a integrated molecular diode-like devices that exhibit reconfigurable behaviour. More specifically, an ionic reservoir can modify the behaviour of the device from a diode to a photodiode. The device behaviour is explained in detail. Nevertheless, the manuscript does not present any computational/processing aspect close or beyond the state-of-art. Therefore, I believe that the manuscript is suitable for a more specialized journal.

Our response:

We would like to thank Reviewer #2 for reading our manuscript. However, we do not “modify the behaviour of the device from a diode to a photodiode”. Instead, we apply rolled-up nanotechnology to develop and integrate new, i.e., process-programmable functional molecular devices on wafer-scale, which can switch between photomultiplication photodiodes and bipolar memristors. The transition depends on the release of mobile ions stored in the bottom polymeric electrode, and can be controlled by modulating the local electric field at the interface between the ultrathin molecular layer and the bottom polymer electrode. From the aspect of processing, it is the first time to realize fully integrated function-switching molecular devices at the sub-10 nm thickness scale by coupling electronic and ionic transport mechanisms. We believe this work goes way beyond state-of-the-art in small-scale organic devices, and we hope it can attract broad interest in the fields of molecular electronics, organic electronics, materials sciences, nanotechnologies and electronics engineering.

Response to the comments of Reviewer #3

Reviewer #3 (Remarks to the Author):

Comments: The manuscript by Li et al. with the title “Process-Programmable Molecular Junctions Switch between Photomultiplication and Memristive Behaviour in On-Chip Integrated Devices” demonstrate memristive behaviour of a new type of molecular junction. They report a new type of molecular electronic device which is very interesting and deserves publication in Nat Commun after the authors have addressed the following points.

Our response:

Thank you very much for your efforts on our manuscript. We greatly appreciate your advice and comments which are very helpful for improving the quality of this work. We have revised the manuscript according to your suggestions. The response and revised parts are marked in blue and red, respectively. Our response is presented point-to-point in the following.

Question (3.1) *“due to the lack of damage-free micro-nano fabrication techniques, it is still a huge challenge to integrate and control the ionic and electronic functions within the molecular devices in parallel at wafer-scale, which is crucial to develop novel and practical functional molecular devices.” This statement is correct for single-molecule junctions, but not for large area junctions. There are quite a few established fabrication techniques that have withstood the test of time , some are even commercially available (DOI: 10.1088/0953-8984/28/9/094011 ; <https://dx.doi.org/10.1021/jacs.9b12424>; but also the groups of Cahen and Lacroix make highly stable molecular junctions on large scales, group of Chiechi reported impressive ways to fabricate functional molecular devices (including memory) with very high stabilities (see doi: 10.1063/5.0050667 for an overview and citations therein). The introduction should be toned down and include a more balanced discussion.*

Response (3.1)

We are very grateful for your suggestion. We agree that it is not proper to skip the well-established fabrication techniques in the introduction. Accordingly, we modified our introduction part and now provide a more balanced discussion. Moreover, we prepared a table (in Supplementary Section 1) to compare the different fabrication techniques.

- 1) In the **Abstract**, we deleted the sentence “However, controlling and manufacturing functional molecular devices are still far from being achievable due to challenges in device construction and

integration.”

- 2) In the **Introduction**, we deleted the sentence “However, due to the lack of damage-free micro-nano fabrication techniques, it is still a huge challenge to integrate and control the ionic and electronic functions within the molecular devices in parallel at wafer-scale, which is crucial to develop novel and practical functional molecular devices.”
- 3) In the **Introduction** (on Page 4, Line 69), we provided a more balanced discussion about different fabrication techniques, as follows:

Molecular junctions are the most investigated sub-10 nm thick molecular devices. Compared to single-molecule junctions, large-area junctions based on molecular ensembles are more promising for scalable fabrication. Various fascinating techniques have been developed to realize ensemble molecular junctions on large scales, such as polydimethylsiloxane (PDMS)-assisted liquid metal contacts¹⁴, suspended nanowire junctions¹⁵, surface-diffusion-mediated deposition¹⁶, nanoskiving¹⁷, carbon paint protective layer-based junctions¹⁸, and electron-beam deposition of carbon¹⁹. A comparison of these techniques for large-area sub-10 nm thick molecular devices is discussed in Supplementary Section 1 and Supplementary Table 1. Among them, rolled-up nanotechnology provides an efficient strategy to fabricate fully integrated functional molecular devices on chip via damage-free soft contacts²⁰, which is not only suitable for ordered self-assembled monolayers but also for deposited ultrathin molecular layers. The latter allows for great flexibility in designing the molecular component and thickness through standard vacuum deposition technologies such as thermal evaporation^{13, 21}.

Related references in the manuscript:

13. Li, T., Bandari, V.K., Hantusch, M., Xin, J., Kuhrt, R., Ravishankar, R., Xu, L., Zhang, J., Knupfer, M., Zhu, F., Yan, D., Schmidt, O.G. Integrated molecular diode as 10 MHz half-wave rectifier based on an organic nanostructure heterojunction. *Nat. Commun.* **11**, 3592 (2020).
14. Karuppanan, S.K., Hongting, H., Troadec, C., Vilan, A., Nijhuis, C.A. Ultrasoft and Photoresist-Free Micropore-Based EGaIn Molecular Junctions: Fabrication and How Roughness Determines Voltage Response. *Adv. Funct. Mater.* **29**, 1904452 (2019).
15. Kayser, B., Fereiro, J.A., Bhattacharyya, R., Cohen, S.R., Vilan, A., Pecht, I., Sheves, M., Cahen, D. Solid-state electron transport via the protein azurin is temperature-independent down to 4 K. *J. Phys. Chem. Lett.* **11**, 144-151 (2019).
16. Bonifas, A.P., McCreery, R.L. ‘Soft’ Au, Pt and Cu contacts for molecular junctions through surface-diffusion-mediated deposition. *Nat. Nanotechnol.* **5**, 612-617 (2010).
17. Pourhossein, P., Chiechi, R.C. Directly addressable sub-3 nm gold nanogaps fabricated by nanoskiving using self-assembled monolayers as templates. *ACS Nano* **6**, 5566-5573 (2012).
18. Karuppanan, S.K., Neoh, E.H.L., Vilan, A., Nijhuis, C.A. Protective layers based on carbon paint to yield high-quality large-area molecular junctions with low contact resistance. *J. Am. Chem. Soc.*

142, 3513-3524 (2020).

19. Bergren, A.J., Zeer-Wanklyn, L., Semple, M., Pekas, N., Szeto, B., McCreery, R.L. Musical molecules: the molecular junction as an active component in audio distortion circuits. *J. Phys. Condens. Matter* **28**, 094011 (2016).
20. Bufon, C.C., Gonzalez, J.D., Thurmer, D.J., Grimm, D., Bauer, M., Schmidt, O.G. Self-assembled ultra-compact energy storage elements based on hybrid nanomembranes. *Nano Lett.* **10**, 2506-2510 (2010).
21. Jalil, A.R., Chang, H., Bandari, V.K., Robaschik, P., Zhang, J., Siles, P.F., Li, G., Burger, D., Grimm, D., Liu, X., Salvan, G., Zahn, D.R., Zhu, F., Wang, H., Yan, D., Schmidt, O.G. Fully Integrated Organic Nanocrystal Diode as High Performance Room Temperature NO₂ Sensor. *Adv. Mater.* **28**, 2971-2977 (2016).

- 4) In *Supplementary Information* (on Page S4), we provide a table to compare different fabrication techniques (see Section 1 and Supplementary Table 1), as follows:

Section 1. Comparison of different techniques for creating sub-10 nm thick molecular devices

A literature study is conducted to compare the rolled-up soft contact technology with other fabrication techniques for creating sub-10 nm thick molecular devices, as shown in Supplementary Table 1. Each technique contributes greatly to the development of molecular electronics^{20, 21}.

Bending induced by releasing built-in strain gradients in thin film layer stacks is an effective approach to self-assemble 2D planar nanomembranes into 3D micro- and nano-tubular architectures, which has provided a smart platform for various fundamental investigations as well as promising applications in the field of the micro- and nanosciences, including magnetics²², biology²³, optics²⁴, robotics²⁵, and energy storage²⁶. At the same time, this self-curling effect can also create a damage-free and self-adjusted top electrode on molecular layers (including self-assembled monolayers, deposited ultrathin molecular layers, and so on) after rolling, hence extending this technique to molecular electronics²⁷. More importantly, the self-rolling process is compatible to conventional photolithography techniques. Therefore, fully integrated sub-10 nm thick molecular devices on wafer scale can be realized by rolled-up soft contacts. Based on this technique some significant achievements have been obtained.

The capability of rolled-up soft contacts is comparable to existing well-established fabrication techniques for creating sub-10 nm thick molecular devices on large scales such as nanoskiving²⁸, surface-diffusion-mediated deposition²⁹, pore-assisted carbon paint³⁰, and so on. Still, there are challenges to tackle when using the rolled-up nanotechnology. First, due to the ultrathin feature of the sacrificial and strained layers, any defect can lead to an uneven stress distribution, resulting in the failure of rolling. Second, the roughness of the top tubular and bottom finger electrode is usually

in the same range of the molecular film thickness. Therefore, the thinner the molecular layer, the more likely the device becomes shorted. Consequently, to obtain a high yield of working devices, much attention should be paid to carefully control the quality and roughness of every film.

Supplementary Table 1 | Comparison of different techniques for creating sub-10 nm thick molecular devices

Techniques	Key principle	Efficiency^a	Structure stability	Addressability^b	Fabrication scalability	Fabrication difficulty	Ref.
Conductive AFM	Metal-coated AFM tip	Low	Low	Difficult	No	Low	31
Crossed wire junction	Lorentz force	Low	Low	Easy	No	Low	32
Liquid metal contact	Surface tension	Low to medium	Low	Difficult	Possible (PDMS channel)	Medium	33, 34
Suspended nanowire junctions	Dielectrophoretic trapping	Medium	High	Difficult	Possible (randomly distributed)	Medium	35, 36
E-beam deposited carbon film	E-beam deposition	Low to medium	High	Easy	High	Low	37, 38
Surface-diffusion-mediated deposition	Atom diffusion	High	High	Difficult	High	High	29
Nanoskiving	Edge lithography	High	High	Easy	High	Medium to high	39
Pore + metal evaporation	Photolithography, metal evaporation	Low	High	Easy	High	High	40
Pore + conductive carbon/polymer	Photolithography, spin coating	High	High	Easy	High	High	30, 41
Lift-and-float approach	Lift-off and floating	Medium	High	Difficult	Possible (polymer-assisted)	Low	42

Metal transfer printing	PDMS Stamp	Medium to high	High	Easy	High (relying on the stamp)	Medium	43
Graphene transfer	Adhesive tape	Medium	High	Easy	Possible	Medium	44
Rolled-up soft contact	Photolithography, stress release	Medium to high	High	Easy	High (full integration)	High	This work

a. Efficiency: Sum of massive fabrication capability and yield.

b. Addressability: Sum of the capability to settle the molecules, to locate the top electrodes, and to measure the molecular junctions.

Related references in the Supplementary Information:

20. Haick, H., Cahen, D. Making contact: Connecting molecules electrically to the macroscopic world. *Prog. Surf. Sci.* **83**, 217-261 (2008).
21. Liu, Y., Qiu, X., Soni, S., Chiechi, R.C. Charge transport through molecular ensembles: Recent progress in molecular electronics. *Chem. Phys. Rev.* **2**, 021303 (2021).
22. Gabler, F., Karnaushenko, D.D., Karnaushenko, D., Schmidt, O.G. Magnetic origami creates high performance micro devices. *Nat. Commun.* **10**, 1-10 (2019).
23. Akbar, F., Rivkin, B., Aziz, A., Becker, C., Karnaushenko, D.D., Medina-Sánchez, M., Karnaushenko, D., Schmidt, O.G. Self-sufficient self-oscillating microsystem driven by low power at low Reynolds numbers. *Sci. Adv.* **7**, eabj0767 (2021).
24. Yin, Y., Wang, J., Wang, X., Li, S., Jorgensen, M.R., Ren, J., Meng, S., Ma, L., Schmidt, O.G. Water nanostructure formation on oxide probed in situ by optical resonances. *Sci. Adv.* **5**, eaax6973 (2019).
25. Bandari, V.K., Nan, Y., Karnaushenko, D., Hong, Y., Sun, B., Striggow, F., Karnaushenko, D.D., Becker, C., Faghieh, M., Medina-Sánchez, M. A flexible microsystem capable of controlled motion and actuation by wireless power transfer. *Nat. Electron.* **3**, 172-180 (2020).

26. Lee, Y., Bandari, V.K., Li, Z., Medina-Sánchez, M., Maitz, M.F., Karnaushenko, D., Tsurkan, M.V., Karnaushenko, D.D., Schmidt, O.G. Nano-biosupercapacitors enable autarkic sensor operation in blood. *Nat. Commun.* **12**, 1-10 (2021).
27. Li, T., Bandari, V.K., Hantusch, M., Xin, J., Kuhrt, R., Ravishankar, R., Xu, L., Zhang, J., Knupfer, M., Zhu, F., Yan, D., Schmidt, O.G. Integrated molecular diode as 10 MHz half-wave rectifier based on an organic nanostructure heterojunction. *Nat. Commun.* **11**, 3592 (2020).
28. Pourhossein, P., Chiechi, R.C. Directly addressable sub-3 nm gold nanogaps fabricated by nanoskiving using self-assembled monolayers as templates. *ACS Nano* **6**, 5566-5573 (2012).
29. Bonifas, A.P., McCreery, R.L. 'Soft' Au, Pt and Cu contacts for molecular junctions through surface-diffusion-mediated deposition. *Nat. Nanotechnol.* **5**, 612-617 (2010).
30. Karuppanan, S.K., Neoh, E.H.L., Vilan, A., Nijhuis, C.A. Protective layers based on carbon paint to yield high-quality large-area molecular junctions with low contact resistance. *J. Am. Chem. Soc.* **142**, 3513-3524 (2020).
31. Hnid, I., Frath, D., Lafalet, F., Sun, X., Lacroix, J.-C. Highly efficient photoswitch in diarylethene-based molecular junctions. *J. Am. Chem. Soc.* **142**, 7732-7736 (2020).
32. Seferos, D.S., Trammell, S.A., Bazan, G.C., Kushmerick, J.G. Probing \$\pi\$ -coupling in molecular junctions. *Proc. Natl Acad. Sci. USA* **102**, 8821-8825 (2005).
33. Nijhuis, C.A., Reus, W.F., Barber, J.R., Dickey, M.D., Whitesides, G.M. Charge transport and rectification in arrays of SAM-based tunneling junctions. *Nano Lett.* **10**, 3611-3619 (2010).
34. Chiechi, R.C., Weiss, E.A., Dickey, M.D., Whitesides, G.M. Eutectic gallium–indium (EGaIn): a moldable liquid metal for electrical characterization of self-assembled monolayers. *Angew. Chem. Int. Ed.* **47**, 142-144 (2008).
35. Kayser, B., Fereiro, J.A., Bhattacharyya, R., Cohen, S.R., Vilan, A., Pecht, I., Sheves, M., Cahen, D. Solid-state electron transport via the protein azurin is temperature-independent down to 4 K. *J. Phys. Chem. Lett.* **11**, 144-151 (2019).

36. Yoon, H.P., Maitani, M.M., Cabarcos, O.M., Cai, L., Mayer, T.S., Allara, D.L. Crossed-nanowire molecular junctions: a new multispectroscopy platform for conduction--structure correlations. *Nano Lett.* **10**, 2897-2902 (2010).
37. Tefashe, U.M., Nguyen, Q.V., Lafalet, F., Lacroix, J.-C., McCreery, R.L. Robust Bipolar Light Emission and Charge Transport in Symmetric Molecular Junctions. *J. Am. Chem. Soc.* **139**, 7436-7439 (2017).
38. Bergren, A.J., Zeer-Wanklyn, L., Semple, M., Pekas, N., Szeto, B., McCreery, R.L. Musical molecules: the molecular junction as an active component in audio distortion circuits. *J. Phys. Condens. Matter* **28**, 094011 (2016).
39. Pourhossein, P., Vijayaraghavan, R.K., Meskers, S.C., Chiechi, R.C. Optical modulation of nano-gap tunnelling junctions comprising self-assembled monolayers of hemicyanine dyes. *Nat. Commun.* **7**, 1-9 (2016).
40. Wang, W., Lee, T., Reed, M.A. Mechanism of electron conduction in self-assembled alkanethiol monolayer devices. *Phys. Rev. B* **68**, 035416 (2003).
41. Akkerman, H.B., Blom, P.W., de Leeuw, D.M., de Boer, B. Towards molecular electronics with large-area molecular junctions. *Nature* **441**, 69-72 (2006).
42. Vilan, A., Cahen, D. Soft contact deposition onto molecularly modified GaAs. Thin metal film flotation: principles and electrical effects. *Adv. Funct. Mater.* **12**, 795-807 (2002).
43. Loo, Y.-L., Lang, D.V., Rogers, J.A., Hsu, J.W. Electrical contacts to molecular layers by nanotransfer printing. *Nano Lett.* **3**, 913-917 (2003).
44. Wang, Z., Dong, H., Li, T., Hviid, R., Zou, Y., Wei, Z., Fu, X., Wang, E., Zhen, Y., Nørgaard, K. Role of redox centre in charge transport investigated by novel self-assembled conjugated polymer molecular junctions. *Nat. Commun.* **6**, 1-10 (2015).

Question (3.2) *The authors state "it is reasonable to claim that the ensembles of CuPc/C₆₀ molecules fall into the molecular scale", but in their explanation of the mechanism of they fall back on semiconductor physics which goes against this statement. Schottky junctions, p-n junctions, band bending, can only form when depletion layers are present (which usually far exceed molecular dimensions), such depletion layers cannot form in molecules. In molecular junctions, HOMO/LUMO levels can shift (energy level alignment) for instance, and since there is only one HOMO/LUMO per molecules, band bending is not possible. The proposed model and explanations suggest that their devices are not truly molecular in nature as ions from the electrode material play a crucial role. The authors should modify their manuscript accordingly*

Response (3.2)

Thank you very much for this valuable comment, and we appreciate it a lot. Our integrated molecular devices are comparable in size to SAM-based junctions (i.e., below 10 nm), however, as pointed out by Reviewer #3 the working principle of our molecular devices falls into the scope of semiconductor physics. Therefore, in order to better represent and describe the dimension and the mechanism, we now term our devices as "sub-10 nm thick molecular devices", in which the classic molecular junctions are also included. We modified our manuscript accordingly, and the main revised points are as follows:

- 1) We changed our **Title** to "On-Chip Integrated Process-Programmable Sub-10 nm Thick Molecular Devices Switching between Photomultiplication and Memristive Behaviour".
- 2) In the **Abstract** (on Page 1, Line 16), we deleted the sentence "However, controlling and manufacturing functional molecular devices are still far from being achievable due to challenges in device construction and integration.", and we rewrote it as follows:

Molecular junctions based on single molecules or self-assembled monolayers are promising candidates for a new generation of nanoelectronic applications as they not only have the potential to transcend Moore's law but also inhere a large range of both physical and chemical functionalities. However, molecular devices constructed by ultrathin molecular layers with dimensions comparable to ensemble molecular junctions, which can offer alternative approaches to realize functional molecular devices, are often overlooked.

- 3) In the **Introduction** (on Page 3, Line 51), we added a description of the meaning of molecular devices constructed by depositing ultrathin molecular layers with dimensions comparable to ensemble molecular junctions, as follows:

Generally, molecular devices focus on molecular junctions, in which the functionalities are realized

by manipulating the chemical properties and electronic structures of the individual or ensemble molecules^{8, 9}. However, the potential of molecular devices constructed by ultrathin molecular layers with dimensions comparable to ensemble molecular junctions has been underestimated. For instance, fundamental studies predicted that short-channel molecular rectifiers can achieve rectification frequencies in the terahertz range^{10, 11}, but a state-of-the-art half-wave SAM rectifier (consisting of Ag^{TS}-SC₁₁Fc₂//Ga₂O₃/EGaIn) was reported to operate at only 50 Hz¹². In marked contrast, Li *et al.* constructed integrated rectifiers based on nanometer-thick hybrid layers consisting of fluorinated cobalt phthalocyanine (F₁₆CoPc)/copper phthalocyanine (CuPc) heterojunctions, which were able to convert alternating current to direct current with a frequency of up to 10 MHz¹³. The excellent high-frequency response was mainly attributed to the band bending of the ultrathin heterojunction. The same dimension but different working principle attracts our special interest to further integrate and control the ionic and electronic functions within ultrathin molecular layer-based devices on the wafer-scale, which is crucial to develop novel and practical functional molecular devices.

Related references in the manuscript:

8. Xin, N., Guan, J., Zhou, C., Chen, X., Gu, C., Li, Y., Ratner, M.A., Nitzan, A., Stoddart, J.F., Guo, X. Concepts in the design and engineering of single-molecule electronic devices. *Nat. Rev. Phys.* **1**, 211-230 (2019).
9. Chen, X., Roemer, M., Yuan, L., Du, W., Thompson, D., Del Barco, E., Nijhuis, C.A. Molecular diodes with rectification ratios exceeding 10⁵ driven by electrostatic interactions. *Nat. Nanotechnol.* **12**, 797-803 (2017).
10. Studel, S., Myny, K., Arkhipov, V., Deibel, C., De Vusser, S., Genoe, J., Heremans, P. 50 MHz rectifier based on an organic diode. *Nat. Mater.* **4**, 597-600 (2005).
11. Celestin, M., Krishnan, S., Bhansali, S., Stefanakos, E., Goswami, D.Y. A review of self-assembled monolayers as potential terahertz frequency tunnel diodes. *Nano Res.* **7**, 589-625 (2014).
12. Nijhuis, C.A., Reus, W.F., Siegel, A.C., Whitesides, G.M. A molecular half-wave rectifier. *J. Am. Chem. Soc.* **133**, 15397-15411 (2011).
13. Li, T., Bandari, V.K., Hantusch, M., Xin, J., Kuhrt, R., Ravishankar, R., Xu, L., Zhang, J., Knupfer, M., Zhu, F., Yan, D., Schmidt, O.G. Integrated molecular diode as 10 MHz half-wave rectifier based on an organic nanostructure heterojunction. *Nat. Commun.* **11**, 3592 (2020).

- 4) We revised the statement “Compared to the size of a single phthalocyanine molecule (~1.5 nm) and a fullerene molecule (~0.7 nm), it is reasonable to claim that the ensembles of CuPc/C₆₀ molecules fall into the molecular scale” to “It is reasonable to claim that the ensembles of CuPc/C₆₀ molecules fall into the sub-10 nm thickness scale” (on Page 6, Line 114).

Question (3.3) *After reading the bold introduction, my expectations were high but ...:*

Response (3.3)

Thank you very much for this comment, and we hope our revision and reply can now meet all your expectations.

Question (3.3.1) *Lack of statistics. No yields, error bars (reproducibilities) etc are reported. Normally heatmaps, tables with statistics (number of samples, number of devices, number of measurements etc), (log)average curves with (log)standard deviations etc should be all given but not reported here.*

Response (3.3.1)

Thank you very much for this suggestion. In the revised version, we provide statistical data of the device yield and performance. We find that the yield increases with the thickness of the molecular hybrid layer, and the values of the molecular devices based on CuPc(3 nm)/C₆₀(3 nm), CuPc(5 nm)/C₆₀(5 nm), and CuPc(20 nm)/C₆₀(20 nm) are 39.7%, 48.5%, and 63.2%, respectively. This is because the roughness (~ 2nm) of the top tubular and bottom finger electrodes is on the same order of the molecular film thickness. Therefore, the thinner the molecular layer, the more likely the device becomes shorted, as shown in Figure R7. On the other hand, our yield is greatly limited by the simple laboratory-grade thermal deposition machine, which brings contamination to the deposited metal films, hence increasing the roughness and causing failure in rolling. The latter can lead to ill-contact with the molecular layer and then open the circuit. Consequently, it is feasible to achieve a much higher yield if we can improve the experimental conditions, such as using industrial-grade equipment.

[Redacted]

Figure R7 | Details of the devices. a, Rolled-up soft-contact illustrated by AFM images. b and c, AFM topography of rolled-up Au tube (measured on the top surface of the tube). The roughness of the Au tube is extracted from the selected area (in the blue dashed box) along the lateral direction of the tube. d, Schematic illustration of the local contacts. The effective gap between the two electrodes shrinks due to the roughness of the Au finger and the Au tube, as indicated by the blue dashed circle. Adapted from our previous work: *Li, T. et al. Integrated molecular diode as 10 MHz half-wave rectifier based on an organic nanostructure heterojunction. Nat. Commun. 11, 3592 (2020).*

The following description has been added in the revised manuscript (on Page 12, Line 235).

Statistics on yield and resistance switching performance are provided in Supplementary Section 9.

The statistics figure is put as Supplementary Figure 11, and the corresponding description has been added in the revised Supplementary Information (on Page S24), as follows:

Section 9. Statistics

Statistics of the molecular devices are shown in Figure R8 (also in Supplementary Fig. 11). As extracted from Figure R8a-c (also in Supplementary Fig. 11a-c), the yields of the molecular devices based on CuPc(3 nm)/C₆₀(3 nm), CuPc(5 nm)/C₆₀(5 nm), and CuPc(20 nm)/C₆₀(20 nm) are 39.7% (=27/68), 48.5% (=33/68), and 63.2% (=43/68), respectively. This trend in yield versus thickness originates from the fact that the thinner the molecular layer, the more likely the device becomes shorted, because the roughness (~ 2nm) of the top tubular and bottom finger electrodes is in the same range of

the molecular film thickness, which can shrink the effective gap between the two electrodes. Statistics of the device performance are also provided in terms of the rectification ratio, ON/OFF ratio, ON-state current, and switching voltage, as shown in Figure R8d-g (also in Supplementary Fig. 11d-g). Since the hysteretic behaviour of PPA60/CuPc(20 nm)/C₆₀(20 nm)/Au is not remarkable, only the parameters of molecular devices based on CuPc(3 nm)/C₆₀(3 nm) and CuPc(5 nm)/C₆₀(5 nm) are presented.

Figure R8 | Statistics. a-c, Distribution maps of the molecular devices based on CuPc(3 nm)/C₆₀(3 nm), CuPc(5 nm)/C₆₀(5 nm), and CuPc(20 nm)/C₆₀(20 nm), respectively. The green and gray squares represent successful and failed devices on the sample substrates, where a “successful device” refers to a device that exhibits I-V hysteresis behaviour and a “failed device” refers to a shorted or open device. d-g, Average values of the rectification ratio at ±0.5 V (d), ON/OFF ratio at ±0.5 V (e), ON-state current at ±0.5 V (f), and the switching voltage (g). Error bars indicate the standard deviation for each parameter of the working devices. “3/3” and “5/5” denote the molecular devices based on CuPc(3 nm)/C₆₀(3 nm) and CuPc(5 nm)/C₆₀(5 nm), respectively. All data are obtained with the bottom finger electrodes grounded. “Forward” and “Reverse” in d indicate the forward diode and reverse diode, respectively. “Forward” and “Reverse” in e-g indicate the positive and negative voltage regions,

respectively.

Question (3.3.2) Stability. No endurance, retention, or demonstration of non-volatility are given.

Response (3.3.2)

Thank you very much for this comment, and we appreciate it a lot. In the revised manuscript, Figure 4h and 4i in the manuscript are updated with the endurance and retention performances (on Page 31), and the corresponding description has been added (on Page 14, Line 290), as follows:

The stability of our molecular memristors was characterized. The retention time of the PPA60/CuPc(3 nm)/C₆₀(3 nm)/Au device is stable and reliable over 2000 s, and the ON/OFF current ratio remains at ~10³ with a continuous reading bias of 0.5 V, indicating good retention stability (Figure R9h, also in Figure 4h in the manuscript). The endurance of the molecular device degrades slightly over the first 100 sweep cycles due to the decrease in the ON-state current, however, the ON/OFF ratio remains in the range of 10² (Figure R9i, also in Figure 4i in the manuscript).

Figure R9 | Molecular bipolar volatile memristors based on PPA60/CuPc (x nm)/C₆₀ (x nm)/Au tube. h, Retention of ON and OFF states of PPA60/CuPc(3 nm)/C₆₀(3 nm)/Au under continuous read-out voltages (at 0.5 V). The device is switched to the ON state by applying a positive voltage pulse at 1.5 V for a few seconds. i, Endurance of ON and OFF states of PPA60/CuPc(3 nm)/C₆₀(3

nm)/Au over the first 100 measurement cycles (read at 0.5 V).

Question (3.3.3) *Fig 2b shows that their technique may be very difficult to scale. I find this not a major issue, but this should be mentioned and compared to other fabrication techniques*

Response (3.3.3)

Thank you very much for this comment. Our fabrication is based on multiple photolithography-deposition-liftoff (or etching) steps, which is the same process used in microfabrication for integration. It is possible to increase to device density by optimizing the circuit design, such as decreasing the size and number of the contact pads for external measurements or arranging the contact pads only around the wafer edges. We admit that although our rolled-up tubes provide smart soft-contacts to protect the ultrathin molecular layers on wafer-scale, their micron-scale dimensions (especially the length and width of the microtubes) limit miniaturization to the nanoscale.

As presented in our **Response (3.1)** to your first **Question**, we modified our introduction part and provided a more balanced discussion. And we also prepared a table to compare different fabrication techniques in Supplementary Section 1 and Supplementary Table 1.

Question (3.3.4) *How well do their devices perform with respect to other devices? For instance, in their ref 7 the on/off ratios are higher, but there are more examples of molecular memristors in the literature*

Response (3.3.4)

Thank you very much for this comment, and we appreciate it a lot. To compare our device performance with respect to other devices, we created a table in Supplementary Section 12 (Supplementary Table 3). There are different ways of classifying memristors, such as according to the material properties, the working mechanisms, the fabrication method, and so on. Here, we would like to focus on bipolar memristors and simply divide them into two categories according to the sequence of the I-V hysteresis loops. For your reading convenience, we put the corresponding Figure R10 below, which is a part of the Supplementary Figure 12 in the revised Supplementary Information file.

Figure R10 | Two types of bipolar memristors according to the I-V hysteresis sequence. a, The I-V hysteresis follows the order of HRS→LRS→LRS→HRS. b, The I-V hysteresis follows the order of HRS→LRS→HRS→LRS.

The following description has been added in the revised manuscript (on Page 14, Line 295).

Compared with previously reported bipolar memristors (Supplementary Section 12), the key parameters (such as storage medium thickness, rectification ratio, ON/OFF ratio, switching voltage, ON-state current density, and so on) of our molecular memristors reach a comparable level except for the endurance stability.

The following description has been added in the revised Supplementary Information (in Section 12, on Page S35).

Section 12. Comparison of the memristive performance with previously reported memristors

Here, we focus on bipolar memristors and divide them into two categories according to the sequence of the I-V hysteresis loops (as shown in Figure R10, also in Supplementary Fig. 12): following the order of (1) HRS→LRS→LRS→HRS (HLLH) or (2) HRS→LRS→HRS→LRS (HLHL), where HRS and LRS are the abbreviations of high resistance state (i.e., OFF state) and low resistance state (i.e., ON state), respectively.

Most HLHL switching phenomena are found in perovskite-based memristors, in which switching is ascribed to the barrier modulation induced by the polarization of the mobile ions in the perovskites or the polarization of the ferroelectric domains. This kind of memristors also exhibits the “switchable diode effect”. Therefore, they can work as both forward and reverse diodes, which depend on the polarization direction. Such HLHL behaviour and switchable diode effect are rarely reported in molecular/organic memristors but they occur in our devices. Compared with previously reported perovskite-based memristors the performance of our devices is comparable in terms of the rectification

ratio, ON/OFF ratio, switching voltage, and retention stability. On the other hand, the ON-state current density of our devices is higher than that of most perovskite-based memristors. Moreover, there is no need to conduct a poling process to activate the memristive switching behaviour in the molecular devices in our work. Poling refers to applying a higher voltage over a period of time to create polarization in perovskite materials. For the molecular devices in our work, it is supposed that the ultrahigh electric field (for example, the average electric field is about 1.67 MV/cm at 1 V) immediately polarizes the mobile ions through the ultrathin molecular hybrid layer. However, compared to the HLLH memristors listed in Supplementary Table 3, the endurance capability of our devices still needs to be enhanced.

Supplementary Table 3 | Comparison of the memristive performance with previously reported memristors

Device structure ^a	Thickness (nm)	Hysteresis order ^b	Forward RR@±V ^c	Reverse RR@±V	OFF/ON ratio@V	Switching voltage (V)	ON-state current density (A/cm ²)	Forming required? ^d	Retention (s)	Endurance (cycles)	Ref.
Ag/SAM/GaO _x /EGaIn	~2	HLLH	—	2.5×10 ⁴ @1.0 V ^e	6.7×10 ³ @-0.3 V	-0.89	~10 ⁻² @-0.5 V	No	1.2×10 ⁴	10 ⁶	79
ITO/Au NPs/ [Ru(L) ₃](PF ₆) ₂ /ITO	10-100	HLLH	—	—	~10 ⁵ @0.3 V	~0.52	~10 ³ @0.3	No	>10 ⁶	~10 ¹²	80
Pt/PA-TsOH/Pt	90	HLLH	—	—	>10 ⁵ @0.1 V	~7.0	—	No	10 ⁵	>10 ³	81
ITO/Poly-1 ⁴⁺ /Al	100	HLLH	—	—	~2.0×10 ² @1.0 V	3.4	~10 ⁻² @1.0	No	1.2×10 ⁴	500	82
Ag/MoS ₂ /MoO _x /Ag	50-100	HLLH	—	—	~10 ⁶ @~0 V	0.1-0.2	~10 @0.01	No	~10 ⁴	3.0×10 ⁴	83
ITO/CsPbBr ₃ ODs/PMMA/Ag	~800	HLLH	—	—	~10 ³ @0.5 V	~1.0	—	Electro-forming	10 ⁵	10 ⁴	84
Pt/BCZT/HAO/Au	50	HLHL	~10 ⁴ @0.1 V	~10 ⁴ @0.1 V	~10 ⁴ @-0.1 V	~0.2	~10 ⁻⁵ @-0.1 V	Not specified	10 ⁴	60	85
FTO/BiFeO ₃ /C-AFM	250	HLHL	~10 ⁴ @2.0 V	~7 @4.0 V	~7.0×10 ³ @2.0 V	~1.0	~10 ⁻² @2.0 V	Poling @±9V	10 ³	—	86
STO/BiFeO ₃ /C-AFM	240	HLHL	—	~100 @1.0 V	~10 ³ @-2.0 V	~1.0	~10 ⁻² @-2.0 V	Poling @±10V	10 ⁶	—	67
STO/SRO/BiFeO ₃ / C-AFM	40	HLHL	~500 @0.6 V	~200 @0.6 V	~753 @0.5 V	~2.0	—	Poling @±3.5V	10 ³	—	59

ITO/PEDOT:PSS/ MAPbI ₃ /Au	300	HLHL	~200 @1.0 V	~300 @1.0 V	>10 ³ @-1.0 V	~1.0	~10 ⁻² @-1.0 V	Poling @±2.5V	—	—	61
SRO/BiFeO ₃ (Ca)/ C-AFM	100	HLHL	—	~10 ³ @10 V	~10 ³ @-2.0 V	~5.0	—	Poling @±12V	—	—	62
PPA60/CuPc/C ₆₀ /Au	6	HLHL	~400 @0.5 V	~900 @0.5 V	~3×10 ³ @0.5 V	~0.8	>10 @0.5 V	No	2000	100	This work

a, The Red part indicates the storage medium.

b, According to the sequence of the I-V hysteresis loops, bipolar memristors are divided into two categories: following the order of (1) HRS→LRS→LRS→HRS (HLLH) or (2) HRS→LRS→HRS→LRS (HLHL), where HRS and LRS are the abbreviations of high resistance state (i.e., OFF state) and low resistance state (i.e., ON state), respectively.

c, RR= Rectification Ratio.

d, The “forming” in filamentary memristors refers to “electroforming”, while in ferroelectric-type memristors it means “poling”.

e, In this work, the rectifying behaviour arises from the extremely asymmetrical I-V hysteresis loops rather than the “switchable diode effect” that usually occurs in ferroelectric-type memristors.

Related references in the Supplementary Information:

59. Hong, S., Choi, T., Jeon, J.H., Kim, Y., Lee, H., Joo, H.Y., Hwang, I., Kim, J.S., Kang, S.O., Kalinin, S.V. Large resistive switching in ferroelectric BiFeO₃ nano-island based switchable diodes. *Adv. Mater.* **25**, 2339-2343 (2013).
61. Xiao, Z., Yuan, Y., Shao, Y., Wang, Q., Dong, Q., Bi, C., Sharma, P., Gruverman, A., Huang, J. Giant switchable photovoltaic effect in organometal trihalide perovskite devices. *Nat. Mater.* **14**, 193-198 (2015).
62. Yang, C.H., Seidel, J., Kim, S.Y., Rossen, P.B., Yu, P., Gajek, M., Chu, Y.H., Martin, L.W., Holcomb, M.B., He, Q., Maksymovych, P., Balke, N., Kalinin, S.V., Baddorf, A.P., Basu, S.R., Scullin, M.L., Ramesh, R. Electric modulation of conduction in multiferroic Ca-doped BiFeO₃ films. *Nat. Mater.* **8**, 485-493 (2009).
67. Jiang, A.Q., Wang, C., Jin, K.J., Liu, X.B., Scott, J.F., Hwang, C.S., Tang, T.A., Lu, H.B., Yang, G.Z. A resistive memory in semiconducting BiFeO₃ thin-film capacitors. *Adv. Mater.* **23**, 1277-1281 (2011).
79. Han, Y., Nickle, C., Zhang, Z., Astier, H., Duffin, T.J., Qi, D., Wang, Z., Del Barco, E., Thompson, D., Nijhuis, C.A. Electric-field-driven dual-functional

molecular switches in tunnel junctions. *Nat. Mater.* **19**, 843-848 (2020).

80. Goswami, S., Matula, A.J., Rath, S.P., Hedstrom, S., Saha, S., Annamalai, M., Sengupta, D., Patra, A., Ghosh, S., Jani, H., Sarkar, S., Motapothula, M.R., Nijhuis, C.A., Martin, J., Goswami, S., Batista, V.S., Venkatesan, T. Robust resistive memory devices using solution-processable metal-coordinated azo aromatics. *Nat. Mater.* **16**, 1216-1224 (2017).
81. Hu, B., Zhu, X., Chen, X., Pan, L., Peng, S., Wu, Y., Shang, J., Liu, G., Yan, Q., Li, R.W. A multilevel memory based on proton-doped polyazomethine with an excellent uniformity in resistive switching. *J. Am. Chem. Soc.* **134**, 17408-17411 (2012).
82. Cui, B.-B., Mao, Z., Chen, Y., Zhong, Y.-W., Yu, G., Zhan, C., Yao, J. Tuning of resistive memory switching in electropolymerized metallopolymeric films. *Chem. Sci.* **6**, 1308-1315 (2015).
83. Bessonov, A.A., Kirikova, M.N., Petukhov, D.I., Allen, M., Ryhanen, T., Bailey, M.J. Layered memristive and memcapacitive switches for printable electronics. *Nat. Mater.* **14**, 199-204 (2015).
84. Yen, M.C., Lee, C.J., Liu, K.H., Peng, Y., Leng, J., Chang, T.H., Chang, C.C., Tamada, K., Lee, Y.J. All-inorganic perovskite quantum dot light-emitting memories. *Nat. Commun.* **12**, 4460 (2021).
85. Silva, J., Silva, J.P.B., Sekhar, K., Pereira, M., Gomes, M. Impact of the ferroelectric layer thickness on the resistive switching characteristics of ferroelectric/dielectric structures. *Appl. Phys. Lett.* **113**, 102904 (2018).
86. Zhang, L., Chen, J., Cao, J., He, D., Xing, X. Large resistive switching and switchable photovoltaic response in ferroelectric doped BiFeO₃-based thin films by chemical solution deposition. *J. Mater. Chem. C* **3**, 4706-4712 (2015).

In all, I find their devices and some of the concepts very interesting (i.e., role of mobile ions) but the authors should refrain from overhyping, provide proper context and acknowledge relevant prior work, and give a balanced discussion regarding the benefit but also disadvantages of their fabrication method before I can support publication.

Our response:

We appreciate these valuable comments and your suggestions on giving a balanced discussion of relevant prior work and different manufacturing techniques, which has helped us to improve the quality of our work. Please refer to our **Response (3.1)** for details.

REVIEWERS' COMMENTS

Reviewer #1 (Remarks to the Author):

Zhu et al have comprehensively addressed the referee concerns; the new material in the main text and SI have significantly improved the quality of the work and the referee-only material is very convincing. Recommend publish as is.

Reviewer #3 (Remarks to the Author):

The authors went through extraordinary efforts to improve their manuscripts. They have addressed all my questions and followed all suggestions which improved their manuscript considerably. Especially the balanced introduction, comparisons to other techniques, and additional information on statistics (and many more) ensure that their work will sever as an important bench mark for future works. I fully support publication now.